# Dynamic TMoE: A Drift-Aware Dynamic Mixture of Experts Framework for Non-Stationary Time Series Forecasting

Jiawen Zhu [1]   Shuhan Liu [2]   Di Weng [1]   Yingcai Wu [2]

## Abstract

Non-stationary time series forecasting is challenged by evolving distribution shifts that static models struggle to capture. While Mixture-of-Experts (MoE) architectures offer a promising paradigm for decoupling complex drift patterns, existing approaches are limited by fixed expert pools and memoryless routing, hampering their ability to adapt to abrupt regime shifts. To address this, we propose **Dynamic TMoE**, a framework that unifies architectural evolution with temporal continuity during learning phase. By detecting distribution shifts via Maximum Mean Discrepancy (MMD), we dynamically instantiate heterogeneous experts and prune redundant ones to optimize capacity. Additionally, a temporal memory router leverages recurrent states and an anomaly repository to ensure stable, context-aware expert selection without requiring test-time updates. Experiments on nine benchmarks demonstrate state-of-the-art performance, reducing MSE by 10.4% and MAE by 7.8%. Code is available at https://github.com/andone-07/Dynamic-TMoE.

## 1. Introduction

Time series forecasting is a cornerstone of critical decision-making systems, ranging from energy management (Salman et al., 2024) and healthcare monitoring (Mishra et al., 2024) to financial analysis (Olorunnimbe & Viktor, 2024). Despite its importance, robust forecasting remains a formidable challenge because real-world signals are inherently non-stationary. As shown in Figure 1(a), these data are characterized by continuous distribution shifts and evolving temporal dependencies (Gama et al., 2014).

In practice, this non-stationarity often manifests as regime shifts, where a single time series switches between distinct statistical distributions. For instance, a system may transition from a stable, periodic seasonal pattern to a highly volatile trend driven by external shocks. Accurately modeling these dynamic transitions remains a fundamental challenge in the development of robust forecasting architectures.

Recent studies mitigate the impact of non-stationarity via distribution adaptation beyond general sequence modeling. The main paradigm involves stationarization. For instance, Non-stationary Transformers (Liu et al., 2022) restore intrinsic non-stationary information into temporal dependencies, while Dish-TS (Fan et al., 2023) utilizes a Dual-CONET method to mitigate shifts between historical and future distributions. Recently, TimeStacker (Liu et al., 2025b) adapts to evolving signal patterns via multi-level observations and frequency-based attention. Despite their efficacy, these monolithic designs exhibit limited flexibility in handling abrupt regime transitions or complex distribution shifts.

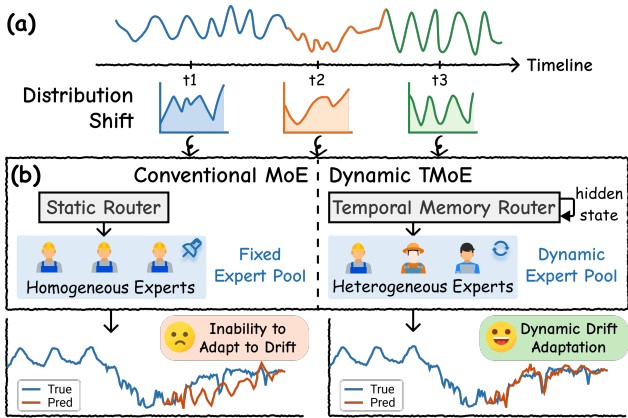

*Figure 1.* **(a)** Time series data with distribution shift. **(b)** Comparison between Conventional MoE and Dynamic TMoE. **Conventional MoE:** Limited by static routing and fixed homogeneous experts, struggling to adapt to distribution shifts. **Dynamic TMoE:** Employs temporal memory routing and a dynamic heterogeneous pool for history-aware, specialized adaptation.

The Mixture-of-Experts (MoE) paradigm provides a promising alternative to monolithic designs by enabling specialized experts to capture the distinct distributions inherent in non-stationary data. For example, TFPS (Sun et al., 2025)

[1]School of Software Technology, Zhejiang University, Ningbo, China [2]State Key Lab of CAD&CG, Zhejiang University, Hangzhou, China. Correspondence to: Di Weng <dweng@zju.edu.cn>.

*Proceedings of the 43rd International Conference on Machine Learning*, Seoul, South Korea. PMLR 306, 2026. Copyright 2026 by the author(s).

leverages subspace clustering to categorize data into distinct patterns, routing them to specialized experts to mitigate pattern drifts. Despite these advancements, existing MoE-based models still encounter two challenges (Figure 1(b)):

**1. Temporal Rigidity.** Conventional MoEs employ fixed expert pools and memoryless routing, constraining their adaptability to evolving distributions. Static expert pools cannot accommodate novel patterns emerging from severe distribution shifts, while memoryless gating ignores temporal continuity, causing erratic and suboptimal adaptation.

**2. Insufficient Specialization.** Conventional MoEs rely on homogeneous experts, lacking the functional diversity required to decouple distinct drift components, forcing the model to redundantly relearn the entire shifted distribution rather than efficiently adapting to specific evolving patterns.

To address these challenges, we propose Dynamic TMoE, a novel framework designed to accommodate non-stationarity via two key mechanisms: (1) A **Temporal Memory Router** coupled with a **Maximum Mean Discrepancy (MMD)-based Evolutionary Strategy** addresses temporal rigidity. The router leverages a Gated Recurrent Unit (GRU) to maintain historical routing context, ensuring temporal continuity in expert selection. The evolutionary strategy monitors distribution shifts via MMD and dynamically instantiates new experts when significant drift is detected. (2) **Heterogeneous experts** address insufficient specialization by employing functionally diverse architectures to capture distinct temporal components (e.g., trend and seasonality), enabling effective disentanglement of complex patterns.

Our main contributions are summarized as follows:

- We introduce a framework that integrates heterogeneous experts with temporal memory-based routing, enabling specialized modeling of distinct temporal patterns while maintaining context-aware, temporally consistent expert selection.
- We propose an MMD-based drift detection mechanism that dynamically expands and prunes the expert pool, allowing model capacity to scale adaptively with distribution shifts.
- We evaluate Dynamic TMoE on nine real-world benchmarks, achieving **state-of-the-art** performance with average reduction of $10.4\%$ in MSE and $7.8\%$ in MAE over competitive baselines. Ablation and case studies further verify our framework's adaptability.

## 2. Related Work

### 2.1. Non-stationary Time Series Forecasting

There are two main categories of approaches to address non-stationary time series forecasting challenges.

**Input-Level Normalization.** Many existing approaches adopt a remove-predict-restore paradigm, aiming to map non-stationary inputs into a stable distribution. RevIN (Kim et al., 2022) pioneered global instance normalization, which SAN (Liu et al., 2023b) later refined by shifting the granularity to local, slice-level adaptation. Moving beyond general-purpose strategies, SIN (Han et al., 2024) focused on the selectivity and interpretability of statistics. Later, FAN (Ye et al., 2024) extended to the frequency domain to address evolving seasonal patterns. Most recently, IN-Flow (Fan et al., 2025) transitioned the field from manual statistic calculation to learnable distribution mapping via invertible flow networks. While effective, these methods often decouple stationarization from the core prediction task, potentially discarding non-stationary signals critical for forecasting.

**Model-Internal Adaptation.** To address the limitations of external normalization, recent studies integrate non-stationary awareness directly into model architectures. One direction focuses on interval-based alignment. For instance, AdaRNN (Du et al., 2021) minimizes distribution mismatches between segments, while Non-stationary Transformers (Liu et al., 2022) and TimeBridge (Liu et al., 2025a) redesign attention mechanisms to restore removed statistics or capture cointegration trends. Another direction leverages physical or spectral dynamics. For instance, Koopa (Liu et al., 2023a) utilizes Koopman operators for disentangling time-variant dynamics and DERITS (Fan et al., 2024) employs frequency derivative transformations to capture evolving spectral patterns. Furthermore, hierarchical frameworks like TimeStacker (Liu et al., 2025b) capture non-stationarity across multiple observational scales.

Despite these advancements, most existing methods rely on a monolithic backbone to handle diverse distribution shifts. This lack of architectural modularity limits the model's ability to specialize in distinct, co-occurring temporal patterns. Thus, we propose Dynamic TMoE to address this gap.

### 2.2. Mixture of Experts for Time Series Forecasting

To address the limitation of monolithic architectures, the Mixture of Experts (MoE) paradigm decomposes complex non-stationary patterns into subproblems handled by specialized experts. Recent advancements have successfully adapted MoE to time series foundation models. Time-MoE (Shi et al., 2025) leverages sparse gating to achieve efficient billion-scale pre-training with reduced inference costs, while Moirai-MoE (Liu et al., 2025c) exploits MoE to enable automatic token-level specialization, explicitly bypassing the limitations of rigid frequency-based heuristics.

Building on these successes, subsequent efforts have focused on refining how experts are assigned to specific data characteristics. For instance, DUET (Qiu et al., 2025) and TFPS (Sun et al., 2025) use temporal or subspace clustering

to route specific data distributions to specialized experts. Similarly, WaveTS (Zhou et al., 2026) incorporates wavelet transforms and channel clustering to manage multi-channel dependencies within the MoE framework.

However, existing MoE architectures typically operate with a static pool of homogeneous experts, which restricts their adaptability to novel or structurally diverse patterns emerging in non-stationary streams. Besides, these models predominantly rely on stateless routing mechanisms that ignore the historical trajectory of the underlying process, causing inconsistent expert selection. In contrast, Dynamic TMoE introduces a heterogeneous expert pool that expands dynamically via drift detection and utilizes a memory-augmented router to preserve temporal continuity across drifts.

## 3. Method

### 3.1. Problem Definition

**Multivariate Time Series Forecasting.** Let $\mathcal{X} = \{\mathbf{x}_t\}_{t=1}^{\infty}$ denote a multivariate time series, where $\mathbf{x}_t \in \mathbb{R}^V$ represents the values of $V$ variables at time step $t$. Given a lookback window of length $L$, denoted as input matrix $\mathbf{X}_t = \mathbf{x}_{t-L+1:t} \in \mathbb{R}^{L \times V}$, the goal of multivariate time series forecasting is to predict the future sequence of length $T$, denoted as $\mathbf{Y}_t = \mathbf{x}_{t+1:t+T} \in \mathbb{R}^{T \times V}$. The objective is to learn a mapping function $\mathcal{F}_\theta$ parameterized by learnable weights $\theta$, to generate the prediction $\hat{\mathbf{Y}}_t = \mathcal{F}_\theta(\mathbf{X}_t)$.

**Non-stationary and Distribution Shift.** Let $P(\mathbf{Y_t}|\mathbf{X_t})$ denote the conditional probability distribution of the data at time $t$. Standard supervised learning assumes a static distribution, i.e., $P(\mathbf{Y_t}|\mathbf{X_t}) = P(\mathbf{Y_{t+\tau}}|\mathbf{X_{t+\tau}})$ for any time lag $\tau$. However, real-world time series are inherently non-stationary. In our setting, we assume the distribution shifts over time, such that $P(\mathbf{Y_t}|\mathbf{X_t}) \neq P(\mathbf{Y_{t+\tau}}|\mathbf{X_{t+\tau}})$.

### 3.2. Overall Architecture

To address the challenges of non-stationarity, we propose Dynamic TMoE. As illustrated in Figure 2, the framework processes patch embeddings through five synergistic components that form a cohesive system of Perception-Decision-Adaptation: (a) **Patch Embedding** partitions the input into overlapping patches and transforms them into high-dimensional patch embeddings. (b) **Distribution Shift Detector** *(Perception)* continuously monitors distributional divergence between historical and current windows, triggering expert pool evolution upon detecting shifts. (c) **Temporal Memory Router** *(Decision)* dynamically routes patches using recurrent memory to synthesize current features with historical context. Upon drift detection, it leverages an anomaly state repository to retrieve prior knowledge, accelerating adaptation to recurring shifts. (d) **Evolvable Expert Manager** *(Adaptation)* dynamically evolves the expert pool

by instantiating new experts for emerging drifts and pruning underutilized ones to ensure structural plasticity. (e) **Heterogeneous Expert Pool** *(Adaptation)* comprises a base expert pool for fundamental patterns and an evolving drift expert pool that expands upon drift detection. Each expert is heterogeneously architected to provide diverse inductive biases, enabling the model to capture multifaceted temporal dynamics and complex multivariate dependencies.

### 3.3. Patch Embedding

We partition the multivariate input series $\mathbf{X} \in \mathbb{R}^{L \times V}$ into overlapping patches of length $P$ and stride $S$. These patches are projected into a latent dimension $D$, generating a sequence of patch embeddings $\mathbf{X}_p \in \mathbb{R}^{N \times D}$, where $N$ is the number of patches. This representation encapsulates local temporal patterns and serves as the primary input for the subsequent modules.

### 3.4. Distribution Shift Detector

This module continuously monitors distributional divergence between historical data and current windows to detect drifts and thus trigger expert pool evolution. We quantify this divergence via Maximum Mean Discrepancy (MMD), a kernel-based metric that measures the distance between distributions in a Reproducing Kernel Hilbert Space. This process is governed by two key components detailed below.

**MMD-based Discrepancy Measurement.** We define a reference window $\mathcal{W}^{\text{ref}} = \{\mathbf{x}_i^{\text{ref}}\}_{i=1}^{N_r}$ for the historical stable distribution and a sliding current window $\mathcal{W}^{\text{cur}} = \{\mathbf{x}_j\}_{j=1}^{N_c}$. The squared MMD distance is computed as:

$$\mathcal{D}_{\text{mmd}}^2(\mathcal{W}^{\text{ref}}, \mathcal{W}^{\text{cur}}) = \frac{1}{N_r^2} \sum_{i,j} k(\mathbf{x}_i^{\text{ref}}, \mathbf{x}_j^{\text{ref}})$$
$$- \frac{2}{N_r N_c} \sum_{i,j} k(\mathbf{x}_i^{\text{ref}}, \mathbf{x}_j) + \frac{1}{N_c^2} \sum_{i,j} k(\mathbf{x}_i, \mathbf{x}_j) \quad (1)$$

where $k(\mathbf{x}_i, \mathbf{x}_j) = \exp(-\|\mathbf{x}_i - \mathbf{x}_j\|^2 / 2\sigma^2)$ is the RBF kernel. The bandwidth $\sigma$ is determined via the median heuristic, set to the median pairwise distance among reference samples to ensure the kernel remains sensitive to current data.

**Adaptive Thresholding.** To account for temporal fluctuations in noise levels, we adopt a dynamic thresholding mechanism rather than a fixed value. We maintain a sliding history buffer $\mathcal{H}$ of the $n$ most recent MMD scores and compute the threshold $\epsilon$ via the $k$-sigma rule:

$$\epsilon = \mu_{\mathcal{H}} + \lambda \cdot \sigma_{\mathcal{H}} \quad (2)$$

where $\mu_{\mathcal{H}}$ and $\sigma_{\mathcal{H}}$ denote the mean and standard deviation of the scores in $\mathcal{H}$, respectively, and $\lambda$ is a hyperparameter controlling detection sensitivity. A drift is identified

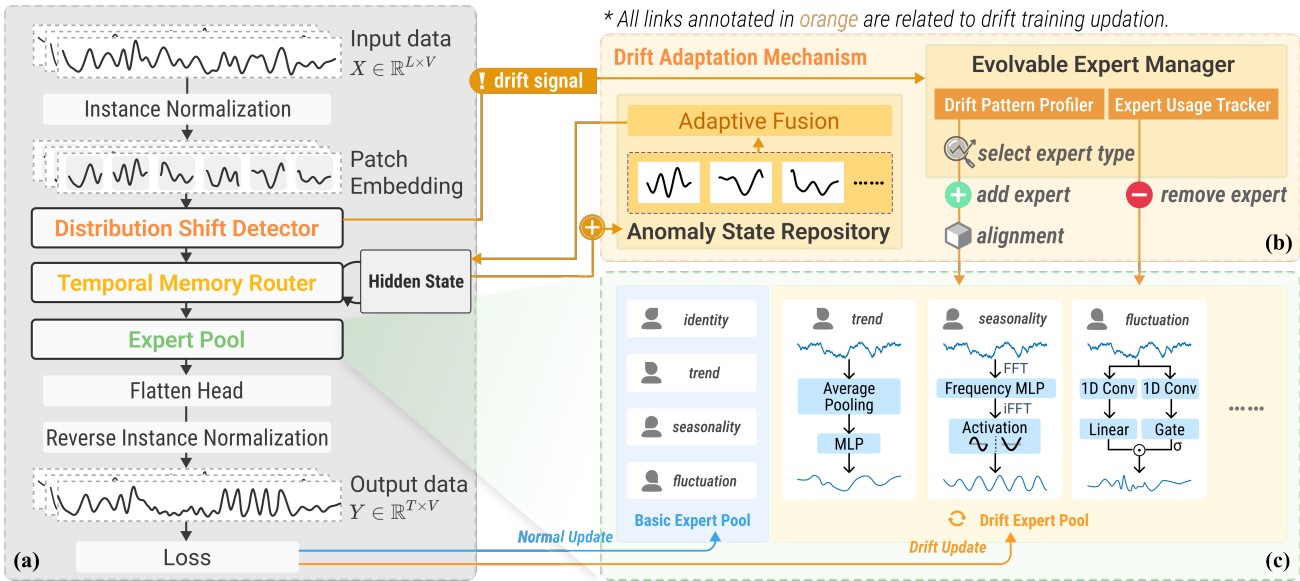

*Figure 2.* Overview of the Dynamic TMoE framework. **(a) Overall Architecture.** Dynamic TMoE employs a closed-loop Perception-Decision-Adaptation mechanism. A Distribution Shift Detector triggers the Evolvable Expert Manager to dynamically add or prune experts in the Drift Expert Pool during training. Simultaneously, the Temporal Memory Router utilizes recurrent hidden states for context-aware routing, leveraging historical states via an Anomaly State Repository and Adaptive Fusion. **(b) Evolvable Expert Manager.** This module utilizes a Drift Pattern Profiler and Expert Usage Tracker to manage capacity. When a shift is detected, it performs expert type selection and pre-training. **(c) Heterogeneous Expert Designs.** Trend Expert extracts global trends via average pooling. Seasonality Expert captures periodic signals in the frequency domain. Fluctuation Expert models local volatility using causal convolutions.

when $\mathcal{D}^2_{\mathrm{mmd}} > \epsilon$, triggering the Evolvable Expert Manager (Sec 3.6). The theoretical analysis in Appendix C proves via a generalization bound that increasing MMD directly elevates the upper bound of the target prediction risk, thereby justifying our MMD-based mechanism.

### 3.5. Temporal Memory Router

This module routes patch embeddings to specialized experts. It addresses the limitations of conventional MoE routers, which process inputs in isolation and neglect temporal dependencies, often leading to erratic expert switching. By integrating historical routing trajectories with temporal context, our approach ensures stable and coherent expert selection. This module comprises three components:

**Sequential State Modeling.** We formulate the routing process as a sequential modeling task to ensure temporal continuity. The router maintains a hidden state $\mathbf{h}_t \in \mathbb{R}^{d_h}$ that encapsulates the evolving temporal dynamics, where $d_h$ represents hidden state dimension. Given the patch embedding $\mathbf{x}_{p,t}$, the state is updated via a GRU:

$$\mathbf{h}_t = \mathrm{GRU}(\phi(\mathbf{x}_{p,t}), \mathbf{h}_{t-1}) \tag{3}$$

where $\phi(\cdot)$ denotes an input projection layer. By conditioning routing decisions on the historical state $\mathbf{h}_{t-1}$ rather than $\mathbf{x}_t$ in isolation, the router treats the input as a coherent sequence. This temporal linkage is designed to mitigate

abrupt expert switching between adjacent patches, aiming to facilitate a routing strategy that evolves smoothly.

**Scalable Top-k Dispatching.** To efficiently accommodate the evolving expert pool $\mathcal{K}_t$ (Sec 3.6), we employ a sparse gating strategy. The router state $\mathbf{h}_t$ is projected to generate logits $\mathbf{l}_t \in \mathbb{R}^{|\mathcal{K}_t|}$. To handle the dynamic expansion of experts, $\mathbf{l}_t$ is composed of concatenated base and drift heads. The final sparse routing weights $\mathbf{g}_t$ are computed as:

$$\mathbf{g}_t = \mathrm{Softmax}(\mathrm{TopK}(\mathbf{l}_t, k)) \tag{4}$$

Thus, only the top-$k$ experts are activated, maintaining a constant $O(k)$ expert computation cost regardless of the pool size.

**Anomaly State Repository.** To accelerate adaptation to recurring drifts, we introduce a memory bank $\mathcal{A}$. Upon drift detection, the router archives the corresponding hidden state into $\mathcal{A}$ to preserve the specialized routing knowledge. During routing, the router synthesizes a context-aware reference prototype $\mathbf{h}_{\mathrm{ref}}$ by retrieving relevant historical states. Specifically, it computes the cosine similarity between the current hidden state $\mathbf{h}_t$ and stored historical states in $\mathcal{A}$, utilizing Softmax normalization to obtain attention weights for aggregation. We employ an adaptive gated fusion mechanism to refine the current state:

$$\tilde{\mathbf{h}}_t = \alpha \cdot \mathbf{h}_t + (1 - \alpha) \cdot \mathbf{h}_{\mathrm{ref}} \tag{5}$$

where $\alpha \in [0, 1]$ is a learnable gating coefficient generated by a fusion network. This mechanism enables the model to dynamically balance between the current observed dynamics and historical routing strategies.

### 3.6. Evolvable Expert Manager

This module governs the adaptation by maintaining the evolution of the expert pool. We treat the expert pool as a dynamic system and implement a comprehensive lifecycle management strategy. Crucially, this structural evolution is confined to the learning stage. This training-time process is realized through three core mechanisms: Drift Pattern Profiler and Post-Addition Alignment focused on informed expansion to address emerging drifts, and Expert Usage Tracker focused on pruning to ensure long-term efficiency.

**Drift Pattern Profiler.** To ensure targeted expansion, we implement a diagnostic mechanism that instantiates experts based on the specific failure mode of the current model. Upon drift detection, a Drift Pattern Profiler identifies the dominant missing patterns by analyzing prediction residuals. Specifically, it computes three disentangled scores to identify the dominant missing pattern: (1) a *Trend Score* based on the goodness-of-fit ($R^2$) of a linear regression over residuals; (2) a *Seasonality Score* derived from the spectral energy concentration in dominant frequencies; and (3) a *Fluctuation Score* measuring the ratio of high-frequency volatility. Detailed implementation is provided in Appendix B.3. A new expert of the matched type is then instantiated and added to the active expert set $\mathcal{K}_t$.

**Post-Addition Alignment.** To ensure the newly instantiated expert bridges the identified knowledge gap without destabilizing the backbone, we implement a targeted alignment phase. We freeze the parameters of existing experts and the router backbone, fine-tuning only the newly added expert and the router's output head on the drift data, which is composed of the concatenated reference ($\mathcal{W}^{\mathrm{ref}}$) and current ($\mathcal{W}^{\mathrm{cur}}$) windows identified by the drift detector. This allows the added expert to settle into the shifted distribution before being integrated into the global optimization loop.

**Expert Usage Tracker.** To maintain an efficient architecture, we implement a pruning mechanism that handles transient or sporadic distribution shifts. Specifically, we track the average routing weight of each expert within a fixed monitoring window. To prevent premature removal due to short-term fluctuations, we enforce a patience constraint where experts are pruned only if they consistently fall below a threshold $\tau$ across $L$ consecutive windows.

### 3.7. Heterogeneous Expert Design

Conventional MoE frameworks typically employ identical expert architectures, which fundamentally restricts their ability to decouple complex temporal dynamics. To address this limitation, we devise a heterogeneous expert pool that incorporates diverse inductive biases tailored to distinct temporal patterns. For notational clarity, let $\mathbf{X}_p \in \mathbb{R}^{N \times D}$ denote the input patch embeddings.

**Identity Expert.** To preserve raw information and ensure stable gradient flow, we include a linear shortcut.

$$E_{\mathrm{id}}(\mathbf{X}_p) = \mathrm{Linear}(\mathbf{X}_p) \tag{6}$$

**Trend Expert.** To extract the global trend of the data while filtering high-frequency noise, we employ average pooling followed by a non-linear projection.

$$E_{\mathrm{trend}}(\mathbf{X}_p) = \mathrm{MLP}(\mathrm{AvgPool}(\mathbf{X}_p)) \tag{7}$$

Here, $\mathrm{AvgPool}(\cdot)$ acts as a low-pass filter, making the expert focus on global trends.

**Seasonality Expert.** To robustly capture periodic patterns amidst temporal noise, we operate in the frequency domain to exploit the inherent sparsity of cyclic signals. We first transform the input to the frequency domain via Fast Fourier Transform (FFT). Then, we employ a Frequency MLP to modulate the spectrum and filter out high-frequency noise. After transforming the features back to the time domain via iFFT, we apply a learnable periodic activation to explicitly model cyclic behaviors:

$$\begin{aligned} \mathbf{Z} &= \mathrm{iFFT}(\mathrm{MLP}(\mathrm{FFT}(\mathbf{X}_p))) \\ E_{\mathrm{sea}}(\mathbf{X}_p) &= \mathrm{Linear}\left(\mathrm{Concat}\left[\sin(\mathbf{Z}_1), \cos(\mathbf{Z}_2)\right]\right) \end{aligned} \tag{8}$$

where $Z$ is split into two halves $\mathbf{Z}_1$ and $\mathbf{Z}_2$ along the channel dimension. This enables the expert to learn a global periodic representation in the frequency domain while refining local cyclic dynamics in the time domain.

**Fluctuation Expert.** To capture high-frequency volatility and sudden local shifts, we utilize 1D causal convolutions equipped with Gated Linear Units.

$$E_{\mathrm{fluc}}(\mathbf{X}_p) = (\mathbf{X}_p * \mathbf{W}_k) \odot \sigma(\mathbf{X}_p * \mathbf{W}_g) \tag{9}$$

where $*$ denotes the convolution operator, $\odot$ is element-wise multiplication, and $\sigma$ is the sigmoid function. This focuses on local receptive fields to model short-term volatility.

**Cyclic Relation Modeling.** To capture evolving variable dependencies, we employ a residual relation learning mechanism. Inspired by CycleNet (Lin et al., 2024), we maintain a learnable periodic prototype $\mathcal{R}_{\mathrm{cycle}} \in \mathbb{R}^{L_{\mathrm{cyc}} \times V \times V}$ to store historical dependency patterns, where $L_{\mathrm{cyc}}$ denotes the cycle length. We derive the instantaneous correlation $\mathcal{R}_{\mathrm{cur}}$ directly from the input features to represent the current variable relationship. Crucially, the final adjacency matrix is reconstructed by adding the learned relation residual to the periodic prototype:

$$\mathcal{R}_{\mathrm{final}} = \mathcal{R}_{\mathrm{cycle}}[t] + \mathrm{MLP}(\mathcal{R}_{\mathrm{cur}} - \mathcal{R}_{\mathrm{cycle}}[t]) \tag{10}$$

*Table 1.* Multivariate time series forecasting results. These are averaged over 4 prediction horizons: $\{24, 36, 48, 60\}$ (ILI benchmark) and $\{96, 192, 336, 720\}$ (All others). For look-back window, we utilize a length of 36 for ILI and 96 for others. Lower MSE and MAE values indicate superior accuracy. The best results are in **bold** and the second best are underlined. Detailed results are in Table 8 of Appendix.

| Models | Dynamic TMoE (Ours) | | TFPS (2025) | | ST-MTM (2025) | | RAFT (2025) | | TimeMixer (2024) | | FITS (2024) | | PatchTST (2023) | | TimesNet (2023) | | DLinear (2023) | | FEDformer (2022) | |
|---|---|---|---|---|---|---|---|---|---|---|---|---|---|---|---|---|---|---|---|---|
| Metric | MSE | MAE | MSE | MAE | MSE | MAE | MSE | MAE | MSE | MAE | MSE | MAE | MSE | MAE | MSE | MAE | MSE | MAE | MSE | MAE |
| Weather | **0.240** | **0.270** | 0.241 | 0.271 | 0.262 | 0.293 | 0.271 | 0.311 | **0.240** | 0.272 | 0.249 | 0.277 | 0.258 | 0.281 | 0.259 | 0.287 | 0.267 | 0.317 | 0.309 | 0.360 |
| Exchange | 0.351 | **0.397** | 0.395 | 0.414 | 0.408 | 0.432 | 0.407 | 0.418 | 0.374 | 0.411 | 0.353 | 0.400 | 0.385 | 0.411 | 0.416 | 0.443 | **0.337** | 0.402 | 0.519 | 0.500 |
| ETTh1 | **0.429** | 0.430 | 0.448 | 0.443 | 0.432 | 0.433 | 0.432 | 0.438 | 0.447 | 0.440 | 0.439 | **0.429** | 0.451 | 0.441 | 0.458 | 0.450 | 0.459 | 0.452 | 0.440 | 0.460 |
| ETTh2 | 0.368 | 0.398 | 0.380 | 0.403 | **0.359** | **0.391** | 0.385 | 0.413 | 0.365 | 0.395 | 0.375 | 0.398 | 0.366 | 0.395 | 0.414 | 0.427 | 0.498 | 0.479 | 0.434 | 0.447 |
| ETTm1 | **0.376** | **0.394** | 0.395 | 0.407 | 0.401 | 0.406 | 0.382 | 0.401 | 0.381 | 0.396 | 0.414 | 0.408 | 0.381 | 0.395 | 0.400 | 0.406 | 0.406 | 0.410 | 0.448 | 0.452 |
| ETTm2 | **0.269** | **0.318** | 0.276 | 0.321 | 0.280 | 0.325 | 0.281 | 0.331 | 0.275 | 0.323 | 0.286 | 0.328 | 0.285 | 0.328 | 0.291 | 0.333 | 0.310 | 0.367 | 0.305 | 0.349 |
| Traffic | 0.479 | **0.288** | **0.457** | 0.304 | 0.556 | 0.337 | 0.537 | 0.347 | 0.485 | 0.298 | 0.627 | 0.377 | 0.488 | 0.309 | 0.620 | 0.336 | 0.625 | 0.385 | 0.610 | 0.376 |
| Electricity | **0.170** | **0.268** | 0.183 | 0.280 | 0.208 | 0.302 | 0.184 | 0.287 | 0.182 | 0.273 | 0.216 | 0.293 | 0.196 | 0.281 | 0.193 | 0.295 | 0.210 | 0.296 | 0.214 | 0.327 |
| ILI | 1.981 | 0.888 | 2.642 | 0.991 | 2.820 | 1.076 | 5.916 | 1.658 | 2.941 | 1.145 | 2.565 | 1.107 | **1.735** | **0.823** | 2.139 | 0.931 | 2.915 | 1.188 | 2.847 | 1.144 |
| #1st | **11** | | 1 | | 2 | | 0 | | 1 | | 1 | | 2 | | 0 | | 1 | | 0 | |

This design leverages stable historical priors while adaptively capturing dynamic shifts through the residual term.

### 3.8. Output Projection

Let $\mathbf{H}^{(L)} \in \mathbb{R}^{B \times V \times N \times D}$ denote the output representations of the final MoE layer, where $B$ is the batch size. First, we perform a flattening operation to merge patch and feature dimensions, yielding a comprehensive feature vector $\mathbf{h}_{b,v} \in \mathbb{R}^{N \cdot D}$ for each variate. Subsequently, a linear projection layer maps this vector directly to the forecasting horizon $T$, generating the final prediction $\hat{\mathbf{Y}} \in \mathbb{R}^{B \times T \times V}$.

## 4. Experiments

### 4.1. Experimental Settings

**Datasets.** To evaluate the performance of Dynamic TMoE, we conducted extensive experiments on nine widely used real-world datasets: ETT (4 subsets) (Zhou et al., 2021), Electricity (Trindade, 2015), Exchange (Lai et al., 2018), Traffic (Lai et al., 2018), Weather (Wu et al., 2021), and Illness (ILI) (Wu et al., 2021), covering domains from energy to healthcare. The detailed description of these datasets is presented in Appendix A.

**Implementation Details.** All experiments are implemented in PyTorch (Paszke et al., 2019) and conducted on a server with two NVIDIA A100 (80G) GPUs. The model is trained using the Mean Squared Error (MSE) loss function and optimized via the AdamW optimizer. The initial learning rate is tuned within the range of $4 \times 10^{-4}$ to $2.4 \times 10^{-3}$ for different datasets, with an early stopping patience of 10 epochs. We use Mean Squared Error (MSE) and Mean Absolute Error (MAE) as evaluation metrics. Following standard protocols, the input look-back window length is $L = 96$ and prediction horizons are $T \in \{96, 192, 336, 720\}$ for most datasets, except ILI which uses $L = 36$ and $T \in \{24, 36, 48, 60\}$.

### 4.2. Main Results

**Baselines.** We compare Dynamic TMoE with nine widely recognized SOTA baselines, including Transformer-based: ST-MTM (Seo & Lim, 2025), PatchTST (Nie et al., 2023), FEDformer (Zhou et al., 2022); CNN-based: TimesNet (Wu et al., 2023); Linear or MLP-based: RAFT (Han et al., 2025), TimeMixer (Wang et al., 2024), FITS (Xu et al., 2024), DLinear (Zeng et al., 2023); We also compare with the latest SOTA MoE-based model: TFPS (Sun et al., 2025).

**Results Analysis.** Dynamic TMoE establishes a new state-of-the-art across a wide range of benchmarks, as detailed in Table 1. Out of 18 evaluation metrics, our model achieves a top-2 ranking in 16 instances (ranking $1^{st}$ in 11 and $2^{nd}$ in 5). The performance is particularly dominant on datasets characterized by high non-stationarity and intricate temporal dependencies, such as Electricity, Weather, and the ETT series. Compared to the leading MoE-based baseline TFPS, our model reduces average MSE by 5.9% and MAE by 3.6%, surpassing it on 8 out of 9 datasets. More broadly, when benchmarking against all SOTA baselines across the nine datasets, Dynamic TMoE reduces the average MSE by 10.4% and MAE by 7.8%. These performance gains validate that our framework has a significant effect in addressing the inherent challenges of non-stationary forecasting.

### 4.3. Ablation Study

We conduct systematic ablations to evaluate the individual contributions of the three core dynamic evolution mechanisms in Dynamic TMoE: (1) the drift-aware adaptation and the temporal memory router, (2) the heterogeneous expert design coupled with relation modeling, and (3) the specific adaptation strategies for drift handling. Detailed specifications for each ablation variant are provided in Appendix F.

**Analysis of dynamic mechanism and temporal memory router.** Table 2 shows that both drift-aware adaptation and temporal consistency are vital for handling non-stationarity. Excluding drift-aware adaptation leads to significant degra-

dation, confirming that static architectures fail to accommodate evolving distributions. Similarly, substituting our GRU-based router with non-temporal alternatives increases MSE by $4.2\%$ and MAE by $1.8\%$ across six benchmarks. This validates that sequential memory is essential for accurate expert assignment in non-stationary environments.

*Table 2.* Ablation study on drift-aware adaptation, temporal memory router, and anomaly memory components. The checkmark ($\checkmark$) indicates the component is included, while ($\times$) indicates it is excluded. Detailed results are provided in Table 10 of the Appendix.

| Model | Drift-Aware Adaptation | Temporal Memory Router | | | Anomaly Memory | ETTh1 | | Weather | |
|---|---|---|---|---|---|---|---|---|---|
| | | GRU | Linear | MLP | | MSE | MAE | MSE | MAE |
| ① | $\checkmark$ | $\checkmark$ | | | $\checkmark$ | **0.429** | **0.430** | **0.240** | **0.270** |
| ② | $\times$ | $\checkmark$ | | | $\checkmark$ | 0.436 | 0.432 | 0.246 | 0.274 |
| ③ | $\checkmark$ | | $\checkmark$ | | $\checkmark$ | 0.436 | 0.430 | 0.247 | 0.275 |
| ④ | $\checkmark$ | | | $\checkmark$ | $\checkmark$ | 0.438 | 0.434 | 0.245 | 0.273 |
| ⑤ | $\checkmark$ | $\checkmark$ | | | $\times$ | 0.438 | 0.434 | 0.246 | 0.274 |

**Analysis of expert diversity and relation modeling.** Table 3 demonstrates that converting our heterogeneous expert pool to a homogeneous one results in a $4.5\%$ increase in MSE and a $2.1\%$ increase in MAE on six benchmarks. This validates that specialized experts more effectively decouple complex distributions for robust generalization. Additionally, removing the cyclic relation layer causes a universal performance drop, leading to a $4.5\%$ increase in MSE and $1.8\%$ in MAE, highlighting the necessity of explicitly modeling inter-variable correlations alongside temporal patterns.

*Table 3.* Ablation study on expert diversity and expert relation layer components. We compare heterogeneous experts against homogeneous variants and evaluate the impact of the relation layer. Detailed results are provided in Table 11 of the Appendix.

| Model | Experts Diversity | | | | | Relation Layer | ETTh1 | | Weather | |
|---|---|---|---|---|---|---|---|---|---|---|
| | Hetero-geneous | All identity | All trend | All seasonality | All fluctuation | | MSE | MAE | MSE | MAE |
| ① | $\checkmark$ | | | | | $\checkmark$ | **0.429** | **0.430** | **0.240** | **0.270** |
| ② | $\checkmark$ | | | | | $\times$ | 0.441 | 0.434 | 0.247 | 0.275 |
| ③ | | $\checkmark$ | | | | $\checkmark$ | 0.440 | 0.434 | 0.246 | 0.274 |
| ④ | | | $\checkmark$ | | | $\checkmark$ | 0.443 | 0.435 | 0.246 | 0.274 |
| ⑤ | | | | $\checkmark$ | | $\checkmark$ | 0.438 | 0.431 | 0.246 | 0.274 |
| ⑥ | | | | | $\checkmark$ | $\checkmark$ | 0.438 | 0.431 | 0.246 | 0.274 |

**Analysis of adaptation strategies.** Table 4 validates our adaptive design choices across the lifecycle of a distribution shift. The MMD-based detector significantly surpasses a simple mean-variance method, which detects drifts by monitoring whether the deviation in mean or variance between the reference and current windows exceeds a threshold, yielding an $8.8\%$ performance gain on the Exchange dataset. Moreover, our Post-Addition Alignment strategy proves superior to directly fine-tuning the base experts, which suffers from catastrophic forgetting, evidenced by a substantial $13.0\%$ MSE increase on ILI when using the latter. Regarding the selection of new expert types, the Drift Pattern Profiler is superior to random selection. Lastly, disabling expert pruning

leads to a decrease in performance, indicating that removing redundant experts helps mitigate overfitting.

*Table 4.* Ablation study on drift detection, new expert alignment strategy, new expert selection, and expert pruning. We compare MMD-based drift detection against simple detection, our Post-Addition Alignment against directly fine-tuning, the Drift Pattern Profiler versus random selection, and the impact of experts pruning. Detailed results are provided in Table 12 of the Appendix.

| Model | Drift Detection | | New Expert Alignment Strategy | | New Expert Selection | | Experts Pruning | Exchange | | ILI | |
|---|---|---|---|---|---|---|---|---|---|---|---|
| | MMD | Simple | Post-Addition Alignment | Fine-tune the Base Experts | Drift Pattern Profiler | Random | | MSE | MAE | MSE | MAE |
| ① | $\checkmark$ | | $\checkmark$ | | $\checkmark$ | | $\checkmark$ | **0.351** | **0.397** | **1.981** | **0.888** |
| ② | | $\checkmark$ | $\checkmark$ | | $\checkmark$ | | $\checkmark$ | 0.382 | 0.413 | 2.082 | 0.904 |
| ③ | $\checkmark$ | | | $\checkmark$ | $\checkmark$ | | $\checkmark$ | 0.351 | 0.397 | 2.238 | 0.938 |
| ④ | $\checkmark$ | | $\checkmark$ | | | $\checkmark$ | $\checkmark$ | 0.365 | 0.404 | 2.260 | 0.952 |
| ⑤ | $\checkmark$ | | $\checkmark$ | | $\checkmark$ | | $\times$ | 0.355 | 0.399 | 2.260 | 0.927 |

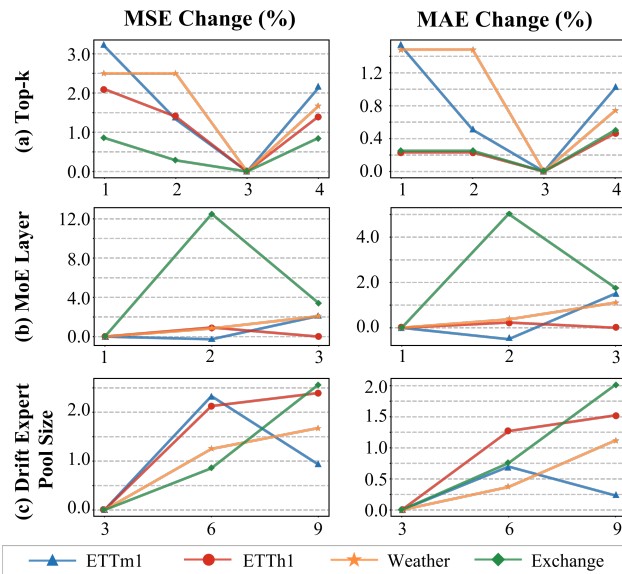

*Figure 3.* Hyperparameter sensitivity analysis results display the relative percentage change in MSE and MAE for: **(a)** the number of activated experts (Top-$k$) relative to the baseline $k = 3$; **(b)** the number of stacked MoE layers relative to the baseline $L = 1$; and **(c)** the drift expert pool size relative to the baseline $S = 3$.

### 4.4. Hyperparameter Sensitivity

To verify the robustness of Dynamic TMoE, we conduct sensitivity analysis on three critical hyperparameters: the number of activated experts (Top-$k$), the number of stacked MoE layers ($L$) and drift expert pool size ($S$).

**Analysis of the number of activated experts.** We vary $k$ within the set $\{1, 2, 3, 4\}$ to investigate the optimal number of activated experts. As illustrated in Figure 3(a), the model consistently achieves the best performance when $k = 3$ across four datasets. While a single expert ($k = 1$) fails to capture complex temporal dynamics, increasing $k$ to 4 leads to degradation, likely due to irrelevant noise. Therefore, $k = 3$ strikes the optimal balance between sparse specialization and cooperative modeling.

*Table 5.* Detailed resource efficiency comparison between Dynamic TMoE and MoE-based baselines (TFPS, FreqMoE, DUET) across nine datasets. We report the number of parameters (M), average inference time per sample (ms/sample), and GPU memory usage (MB).

| Dataset | Dynamic TMoE | | | TFPS | | | FreqMoE | | | DUET | | |
|---|---|---|---|---|---|---|---|---|---|---|---|---|
| | Param (M) | Time (ms) | Mem (MB) | Param (M) | Time (ms) | Mem (MB) | Param (M) | Time (ms) | Mem (MB) | Param (M) | Time (ms) | Mem (MB) |
| **ETTh1** | 8.16 | 0.2783 | 51.15 | 11.16 | 0.1096 | 89.42 | 0.22 | 0.8795 | 26.01 | 3.78 | 0.8402 | 1445.17 |
| **ETTh2** | 9.33 | 0.4336 | 55.23 | 10.76 | 0.0955 | 86.26 | 0.22 | 0.7752 | 26.01 | 3.78 | 0.8611 | 1446.52 |
| **ETTm1** | 3.72 | 0.1750 | 31.98 | 219.48 | 0.2133 | 1559.69 | 0.22 | 0.2127 | 26.37 | 3.98 | 0.9322 | 1498.76 |
| **ETTm2** | 7.83 | 0.1833 | 49.52 | 103.79 | 0.1398 | 673.99 | 0.22 | 0.2238 | 26.37 | 0.45 | 0.8609 | 1486.81 |
| **Weather** | 9.40 | 0.1215 | 65.94 | 43.21 | 0.1497 | 354.00 | 0.22 | 0.2584 | 55.59 | 4.83 | 0.9759 | 1612.77 |
| **ECL** | 24.01 | 3.2829 | 388.79 | 898.38 | 2.7727 | 7052.74 | 0.22 | 0.6436 | 686.49 | 4.83 | 2.9624 | 3039.94 |
| **Exchange** | 13.81 | 0.8655 | 74.65 | 12.23 | 0.3646 | 108.25 | 0.22 | 2.2163 | 27.81 | 27.89 | 0.8535 | 1434.66 |
| **ILI** | 115.40 | 5.1590 | 466.24 | 1.01 | 2.7145 | 16.80 | 0.01 | 13.5581 | 11.24 | 1.28 | 3.5924 | 1398.28 |
| **Traffic** | 128.69 | 8.6058 | 1218.54 | 2360.83 | 4.6406 | 18547.39 | 0.22 | 1.0597 | 1901.79 | 9.04 | 12.6215 | 18024.85 |

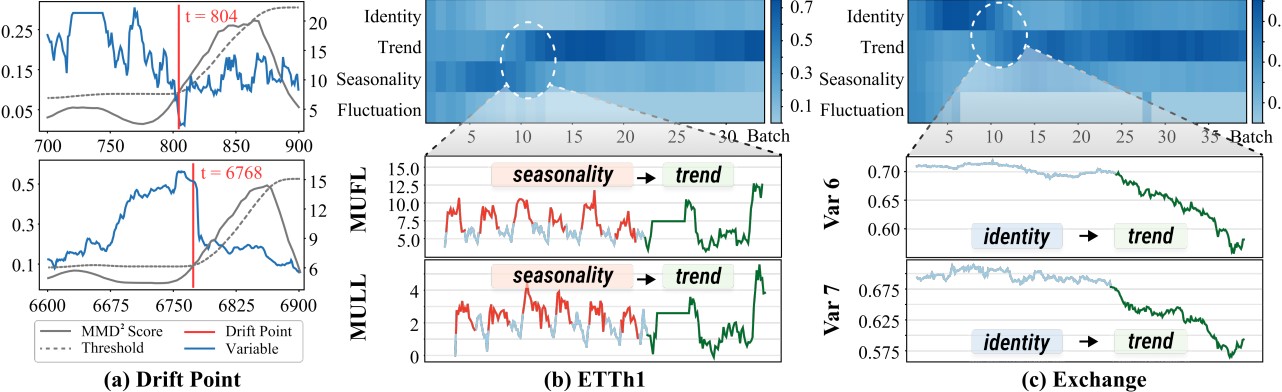

**(a) Drift Point**    **(b) ETTh1**    **(c) Exchange**

*Figure 4.* Visualization of dynamic mechanism. **(a)** Raw series and MMD scores on ETTh1 and Weather. The MMD scores spike and cross dynamic thresholds exactly at drift points, triggering adaptation. **(b)** Router weight heatmaps and corresponding raw series on ETTh1. As the data pattern evolves, the router dynamically allocates dominance to the semantically matching expert. **(c)** Router weight heatmaps and corresponding raw series on Exchange.

**Analysis of the number of stacked MoE layers.** Varying the stacked MoE layers from 1 to 3 (Figure 3(b)) reveals a general preference for shallower structures across most datasets. While ETTm1 exhibits a marginal performance gain with 2 layers, Exchange and Weather perform best with a single layer. Notably, increasing depth on Exchange results in significant performance degradation, indicating that deeper models are prone to overfitting. Consequently, we adopt a 1 or 2-layer structure as our default configuration.

**Analysis of drift expert pool size.** Lastly, we examine the impact of the drift expert pool size by varying it within {3, 6, 9}. As shown in Figure 3(c), a compact pool of 3 experts achieves the best performance. Enlarging the pool size to 6 or 9 leads to general performance degradation. This indicates that larger pools introduce unnecessary complexity and overfitting risks.

### 4.5. Efficiency Analysis

**Results.** To comprehensively assess resource efficiency, we evaluate Dynamic TMoE against three recent MoE-based baselines: TFPS, FreqMoE (Liu, 2025), and DUET (Qiu et al., 2025). As detailed in Table 5, Dynamic TMoE offers a competitive balance between computational cost and ar-

chitectural capacity. While FreqMoE features a lightweight parameter count, its memory consumption scales poorly on datasets with complex temporal dynamics, such as ECL (686.49 MB) and Traffic (1901.79 MB). In contrast, Dynamic TMoE effectively avoids the parameter explosion observed in TFPS and the severe memory bottleneck inherent to DUET. Overall, Dynamic TMoE emerges as the most memory-efficient model on average and achieves a faster average inference time compared to both FreqMoE and DUET. These results highlight that our framework scales elegantly without sacrificing inference speed or memory efficiency.

**Latency Trade-off.** Dynamic TMoE exhibits higher end-to-end inference latency than TFPS, which reflects an architectural trade-off. Conventional MoE frameworks typically employ memoryless and stateless routing mechanisms that are highly parallelizable and computationally fast. However, such static routing ignores historical context and frequently results in unstable or suboptimal expert selection when facing non-stationary data streams. In contrast, our Temporal Memory Router treats the routing process as a sequential modeling task and utilizes a recurrent structure to ensure coherent expert assignment. The inherent sequential dependencies of this recurrent mechanism limit the potential for massive parallelization, thereby introducing the observed

latency overhead.

### 4.6. Case Study

To validate the interpretability and efficacy of Dynamic TMoE, we visualize the internal decision-making process.

Figure 4(a) validates the reliability of our drift detector. On both datasets, the drift detector demonstrates high sensitivity to the onset of distribution shifts. This prompt and precise detection is crucial for our dynamic mechanism, ensuring that the drift expert pool and model adaptation are triggered exactly when non-stationarity emerges.

Figure 4(b)-(c) provides dual validation for both the temporal memory router and the semantic rationality of our heterogeneous experts. Taking ETTh1 (b) as an example, when the data characteristics physically transition from stable high-frequency cycles to a relatively obvious trend pattern, the model autonomously shifts the dominant weights from the seasonality expert to the trend expert. This alignment validates that our experts genuinely specialize in distinct physical patterns, while the router accurately adapts to statistical shifts. Consequently, the framework effectively tackles the challenge of distribution shifts by leveraging the synergy between specialized representation learning and dynamic architectural adaptation.

## 5. Conclusion

In this paper, we presented Dynamic TMoE, a drift-aware framework that overcomes the structural rigidity of conventional MoE architectures by treating the expert pool as an evolvable system. By integrating MMD-based drift detection with a heterogeneous expert pool, our model dynamically expands its capacity to accommodate emerging distribution shifts. Furthermore, the proposed temporal memory router resolves inconsistent gating by employing a recurrent mechanism to maintain temporal continuity and an anomaly repository to retrieve stored hidden states. Empirical results confirm that Dynamic TMoE significantly outperforms state-of-the-art baselines. Future work will explore test-time adaptation to enable continuous expert evolution during inference without offline re-training.

### Acknowledgements

This work was supported by Zhejiang Provincial Natural Science Foundation of China (LD25F020003), NSFC (62421003, 62402421), and Ningbo Yongjiang Talent Programme (2024A-399-G).

## Impact Statement

This paper presents work whose goal is to advance the field of machine learning. There are many potential societal consequences of our work, none of which we feel must be specifically highlighted here.

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

# A. Datasets

To verify the performance of Dynamic TMoE under various distribution shift scenarios, we utilized nine real-world benchmark datasets. These datasets represent a wide range of domains, sampling frequencies, and time series characteristics.

**ETT (Electricity Transformer Temperature)** (Zhou et al., 2021) The ETT dataset is a crucial benchmark in the energy sector, collected from two electricity transformers over a period of two years (July 2016 to July 2018). The dataset comprises *four subsets* based on different sampling intervals. ETTh1 and ETTh2 are recorded at an hourly frequency. ETTm1 and ETTm2 are recorded at a 15-minute frequency.

**ECL (Electricity)** (Trindade, 2015) This dataset contains the hourly electricity consumption of 321 clients, recorded from 2012 to 2014. The data reflects distinct daily and weekly patterns typical of user consumption behavior.

**Exchange** (Lai et al., 2018) The Exchange dataset comprises the daily exchange rates of eight distinct countries (Australia, the United Kingdom, Canada, Switzerland, China, Japan, New Zealand, and Singapore) against the US dollar, collected from 1990 to 2016. This dataset is characterized by non-periodicity and high volatility due to economic fluctuations.

**Traffic** (Lai et al., 2018) This dataset describes the road occupancy rates measured by 862 sensors on San Francisco Bay Area freeways. The data is sampled hourly from 2015 to 2016. It exhibits clear spatial dependencies and circadian rhythms but also contains irregularities caused by traffic events.

**Weather** (Wu et al., 2021) Recorded by the Max Planck Institute for Biogeochemistry, this dataset includes 21 meteorological indicators collected every 10 minutes throughout 2020.

**ILI (Illness)** (Wu et al., 2021) Collected by the Centers for Disease Control and Prevention (CDC) of the United States, this dataset records the weekly ratio of patients with influenza-like illness to the total number of patients from 2002 to 2021. It features a small sample size and strong but shifting cyclic patterns corresponding to flu seasons.

Details for each dataset are provided in Table 6.

*Table 6.* Detailed statistical information of the nine real-world benchmark datasets used for evaluation. The table summarizes the multivariate dimensions, sampling frequency, prediction length, and dataset split information. Among them, Dataset Split denotes the total number of time points in (Train, Validation, Test) split respectively.

| Dataset | Variables | Frequency | Dataset Split | Prediction Length | Domain |
|---|---|---|---|---|---|
| ETTh1 | 7 | 1 Hour | (8545, 2881, 2881) | {96, 192, 336, 720} | Energy |
| ETTh2 | 7 | 1 Hour | (8545, 2881, 2881) | {96, 192, 336, 720} | Energy |
| ETTm1 | 7 | 15 Minutes | (34465, 11521, 11521) | {96, 192, 336, 720} | Energy |
| ETTm2 | 7 | 15 Minutes | (34465, 11521, 11521) | {96, 192, 336, 720} | Energy |
| Traffic | 862 | 1 Hour | (12185, 1757, 3509) | {96, 192, 336, 720} | Transportation |
| Weather | 21 | 10 Minutes | (36792, 5271, 10540) | {96, 192, 336, 720} | Climatology |
| Electricity | 321 | 1 Hour | (18317, 2633, 5261) | {96, 192, 336, 720} | Energy |
| Exchange | 8 | 1 Day | (5215, 761, 1518) | {96, 192, 336, 720} | Economics |
| Illness | 7 | 1 Week | (653, 110, 206) | {24, 36, 48, 60} | Healthcare |

# B. Implementation Details

## B.1. Evaluation Metrics

To rigorously evaluate the forecasting performance of Dynamic TMoE, we use two conventional metrics Mean Squared Error (MSE) and Mean Absolute Error (MAE) on normalized data. Given the ground truth $\mathbf{Y} = \{y_i\}_{i=1}^{N}$ and corresponding predictions $\hat{\mathbf{Y}} = \{\hat{y}_i\}_{i=1}^{N}$, where $N$ denotes the total number of data points across the prediction horizon and test set, the metrics are defined as:

$$\text{MSE} = \frac{1}{N} \sum_{i=1}^{N} (y_i - \hat{y}_i)^2, \quad \text{MAE} = \frac{1}{N} \sum_{i=1}^{N} |y_i - \hat{y}_i|$$

Lower values for both metrics indicate better model performance.

### B.2. Detailed Experiment Settings

All experiments are implemented in PyTorch (Paszke et al., 2019) and conducted on a server equipped with two NVIDIA A100 (80G) GPUs. The model is optimized using the AdamW optimizer with the Mean Squared Error (MSE) loss function, while performance is evaluated using both MSE and MAE. Following standard benchmarks, we fix the look-back window length $L = 96$ for all datasets except ILI ($L = 36$), and select prediction horizons $T$ from $\{96, 192, 336, 720\}$ ($\{24, 36, 48, 60\}$ for ILI). By default, the Dynamic TMoE is stacked with 1 or 2 layers, utilizing a 1 or 2-layer RNN for the router with the top-$k$ set to 3. For the drift adaptation mechanism, the MMD detector adopts a dynamic threshold with a coefficient $\lambda = 3.0$. The batch size is set between 32 and 256. We perform a grid search to determine dataset-specific hyperparameters. The initial learning rate is tuned within the range of $4 \times 10^{-4}$ to $2.4 \times 10^{-3}$ for different datasets. Detailed model configurations, including patch length $P$, stride $S$, and specific learning rates for each dataset, are summarized in Table 7.

### B.3. Drift Pattern Profiler Implementation

To ensure the Evolvable Expert Manager instantiates the most appropriate expert type upon drift detection, we implement a Drift Pattern Profiler. This module operates on the learning discrepancy between the target ground truth $G$ and the current model's output $\hat{G}_{base}$. Let $E \in \mathbb{R}^{N \times P \times D}$ denote the residual error, where $E = G - \hat{G}_{base}$. We first normalize $E$ to zero mean and unit variance per sample to ensure scale invariance:

$$\tilde{E} = \frac{E - \mu_E}{\sigma_E + \epsilon} \tag{11}$$

The Profiler then computes three distinct scores: $S_{trend}$, $S_{sea}$, and $S_{fluc}$ to categorize the drift into one of three prototypes.

**Trend Score ($S_{trend}$).** To assess whether the drift is driven by a shift in the global trend, we fit a linear regression model to the normalized residuals $\tilde{E}$ over the time axis $t \in [-1, 1]$. We calculate the coefficient of determination ($R^2$) to measure how well the linear trend explains the residual variance:

$$S_{trend} = \text{Avg}\left(\max\left(0, 1 - \frac{\sum(\tilde{E} - \tilde{E}_{fit})^2}{\sum \tilde{E}^2}\right)\right) \tag{12}$$

where $\tilde{E}_{fit}$ represents the fitted linear values. A high $S_{trend}$ indicates that the model is failing to capture a clear directional shift.

**Seasonality Score ($S_{sea}$).** To detect emerging periodic patterns, we analyze the residuals in the frequency domain. We compute the power spectrum via the Fast Fourier Transform (FFT): $\mathcal{P} = |\text{FFT}(\tilde{E})|^2$. The seasonality score is defined as the concentration of energy in the top-$k$ dominant frequencies:

$$S_{sea} = \text{Avg}\left(\frac{\sum_{f \in \text{Top-k}(\mathcal{P})} \mathcal{P}(f)}{\sum_f \mathcal{P}(f)}\right) \tag{13}$$

In our implementation, we set $k = 3$. A high $S_{sea}$ implies that the residuals contain significant periodic components that the current experts are missing.

**Fluctuation Score ($S_{fluc}$).** To identify high-frequency volatility or noise-like drifts, we calculate the ratio of energy present in the high-frequency band:

$$S_{fluc} = \text{Avg}\left(\frac{\sum_{f > f_{Nyq}/2} \mathcal{P}(f)}{\sum_f \mathcal{P}(f)}\right) \tag{14}$$

where $f_{Nyq}$ is the Nyquist frequency. This score captures rapid, local variations.

Finally, we compare the normalized scores ($S_{trend}$, $S_{sea}$, $S_{fluc}$) to identify the most significant component of the distribution shift. This ensures that the instantiated expert addresses the most fundamental aspect of the non-stationarity.

## C. Theoretical Analysis

In this section, we provide a rigorous theoretical justification for the Distribution Shift Detector. We establish a generalization error bound based on the Maximum Mean Discrepancy (MMD), demonstrating that controlling the MMD distance between historical and current windows effectively bounds the potential risk on the target distribution.

## C.1. Preliminaries and Definitions

Let $\mathcal{X}$ denote the input space and $\mathcal{Y}$ denote the output space. We consider a domain to be defined by a joint distribution $\mathcal{D}$ over $\mathcal{X} \times \mathcal{Y}$.

- **Source Domain (Reference Window)**: Let $\mathcal{D}_S$ denote the joint distribution of the reference window $W_{\text{ref}}$. We denote its marginal distribution over $\mathcal{X}$ as $P_S$.

- **Target Domain (Current Window)**: Let $\mathcal{D}_T$ denote the joint distribution of the current window $W_{\text{cur}}$. We denote its marginal distribution over $\mathcal{X}$ as $P_T$.

Let $h : \mathcal{X} \to \mathcal{Y}$ be a hypothesis from a hypothesis class $\mathcal{H}$. We define the expected risk of $h$ on distribution $\mathcal{D}$ as:

$$\epsilon_{\mathcal{D}}(h) = \mathbb{E}_{(x,y)\sim\mathcal{D}}[\ell(h(x), y)] \tag{15}$$

where $\ell : \mathcal{Y} \times \mathcal{Y} \to \mathbb{R}_{\geq 0}$ is a non-negative loss function.

**Definition C.1 (Maximum Mean Discrepancy).** Let $k : \mathcal{X} \times \mathcal{X} \to \mathbb{R}$ be a characteristic kernel with associated Reproducing Kernel Hilbert Space (RKHS) $\mathcal{H}_K$ and feature map $\phi : \mathcal{X} \to \mathcal{H}_K$ such that $k(x, x') = \langle \phi(x), \phi(x') \rangle_{\mathcal{H}_K}$. The MMD between two probability distributions $P$ and $Q$ over $\mathcal{X}$ is defined as:

$$\text{MMD}(P, Q) = \|\mathbb{E}_{x\sim P}[\phi(x)] - \mathbb{E}_{x\sim Q}[\phi(x)]\|_{\mathcal{H}_K} \tag{16}$$

Equivalently, this can be expressed as:

$$\text{MMD}(P, Q) = \sup_{f\in\mathcal{H}_K, \|f\|_{\mathcal{H}_K}\leq 1} |\mathbb{E}_{x\sim P}[f(x)] - \mathbb{E}_{x\sim Q}[f(x)]| \tag{17}$$

## C.2. Assumptions

To establish our generalization bound, we introduce the following assumptions.

**Assumption C.2 (Covariate Shift).** The conditional distribution of $Y$ given $X$ remains invariant across the source and target domains:

$$P_S(Y|X) = P_T(Y|X) =: P(Y|X) \tag{18}$$

This assumption states that while the marginal distribution of inputs may shift over time, the underlying functional relationship between inputs and outputs remains stable. This is a standard assumption in domain adaptation literature (Pan & Yang, 2010) and is appropriate for time series forecasting (Du et al., 2021).

**Assumption C.3 (Bounded RKHS Norm).** There exists a constant $L > 0$ such that for any hypothesis $h \in \mathcal{H}$, the conditional expected loss function

$$g_h(x) := \mathbb{E}_{y\sim P(Y|X=x)}[\ell(h(x), y)] \tag{19}$$

belongs to the RKHS $\mathcal{H}_K$ and satisfies:

$$\|g_h\|_{\mathcal{H}_K} \leq L \tag{20}$$

## C.3. Finite Sample Analysis

In practice, we calculate the discrepancy using finite samples from the source and target distributions. Therefore, before analyzing the risk, we must first establish that this empirical calculation is a reliable proxy for the true distribution distance. The following theory establishes the concentration of the empirical MMD estimator.

**Theorem C.4 (Empirical MMD Concentration).** *Let $\{x_i\}_{i=1}^{N_s} \overset{\text{i.i.d.}}{\sim} P_S$ and $\{x'_j\}_{j=1}^{N_t} \overset{\text{i.i.d.}}{\sim} P_T$ be independent samples. Assume the kernel $k$ is bounded: $k(x, x) \leq K$ for all $x \in \mathcal{X}$. Then, with probability at least $1 - \delta$:*

$$\left|\widehat{\text{MMD}} - \text{MMD}(P_S, P_T)\right| \leq 2\sqrt{K}\left(\frac{1}{\sqrt{N_s}} + \frac{1}{\sqrt{N_t}}\right) + \sqrt{\frac{2K\log(2/\delta)}{\min(N_s, N_t)}} \tag{21}$$

*where $\widehat{\text{MMD}}$ denotes the unbiased empirical estimator of MMD.*

**Proof Sketch.** This result follows from standard concentration inequalities for U-statistics. The empirical MMD is a two-sample U-statistic, and its concentration around the population MMD can be established using McDiarmid's inequality combined with bounds on the variance of U-statistics (Gretton et al., 2012).

This theorem provides theoretical support for the reliability of the Perception phase of our framework. It proves that the MMD score calculated by our module converges to the true population discrepancy as the window sizes $N_s$ and $N_t$ allow. This justifies our design choice of using sliding windows to monitor non-stationarity. Despite being a sampling-based approximation, the calculated score $\mathcal{D}^2_{mmd}$ effectively captures meaningful distributional shifts in the time series.

### C.4. Generalization Bound

**Theorem C.5** (**Generalization Bound via MMD**). *Under Assumptions C.2 and C.3, for any hypothesis $h \in \mathcal{H}$, the expected risk on the target domain is bounded by:*

$$\epsilon_{\mathcal{D}_T}(h) \leq \epsilon_{\mathcal{D}_S}(h) + L \cdot \mathrm{MMD}(P_S, P_T) \tag{22}$$

**Proof.** We proceed in three steps.

**Step 1: Decomposition under Covariate Shift.** Under Assumption C.2, the expected risk on any domain $\mathcal{D}$ with marginal $P$ can be written as:

$$\epsilon_{\mathcal{D}}(h) = \mathbb{E}_{x \sim P}\left[\mathbb{E}_{y \sim P(Y|X=x)}[\ell(h(x), y)]\right] \tag{23}$$

Define the conditional expected loss function $g_h : \mathcal{X} \to \mathbb{R}$ as:

$$g_h(x) := \mathbb{E}_{y \sim P(Y|X=x)}[\ell(h(x), y)] \tag{24}$$

Since $P(Y|X)$ is shared between domains (Assumption C.2), the function $g_h$ is identical for both source and target. Thus:

$$\epsilon_{\mathcal{D}_S}(h) = \mathbb{E}_{x \sim P_S}[g_h(x)], \quad \epsilon_{\mathcal{D}_T}(h) = \mathbb{E}_{x \sim P_T}[g_h(x)] \tag{25}$$

**Step 2: Normalization.** By Assumption C.3, we have $g_h \in \mathcal{H}_K$ with $\|g_h\|_{\mathcal{H}_K} \leq L$. Define the normalized function:

$$\tilde{g}_h := \frac{g_h}{L} \tag{26}$$

Then $\tilde{g}_h \in \mathcal{H}_K$ and satisfies:

$$\|\tilde{g}_h\|_{\mathcal{H}_K} = \frac{\|g_h\|_{\mathcal{H}_K}}{L} \leq 1 \tag{27}$$

Thus, $\tilde{g}_h$ lies in the unit ball of the RKHS $\mathcal{H}_K$.

**Step 3: Applying the MMD Bound.** By the dual characterization of MMD (Definition C.1), for any function $f \in \mathcal{H}_K$ with $\|f\|_{\mathcal{H}_K} \leq 1$:

$$|\mathbb{E}_{x \sim P_T}[f(x)] - \mathbb{E}_{x \sim P_S}[f(x)]| \leq \mathrm{MMD}(P_S, P_T) \tag{28}$$

Since $\tilde{g}_h$ satisfies $\|\tilde{g}_h\|_{\mathcal{H}_K} \leq 1$, we can apply this inequality to obtain:

$$|\mathbb{E}_{x \sim P_T}[\tilde{g}_h(x)] - \mathbb{E}_{x \sim P_S}[\tilde{g}_h(x)]| \leq \mathrm{MMD}(P_S, P_T) \tag{29}$$

Substituting $\tilde{g}_h = g_h/L$ and multiplying both sides by $L$:

$$|\mathbb{E}_{x \sim P_T}[g_h(x)] - \mathbb{E}_{x \sim P_S}[g_h(x)]| \leq L \cdot \mathrm{MMD}(P_S, P_T) \tag{30}$$

Using the results from Step 1:

$$|\epsilon_{\mathcal{D}_T}(h) - \epsilon_{\mathcal{D}_S}(h)| \leq L \cdot \mathrm{MMD}(P_S, P_T) \tag{31}$$

Since we seek an upper bound on $\epsilon_{\mathcal{D}_T}(h)$, we have:

$$\epsilon_{\mathcal{D}_T}(h) \leq \epsilon_{\mathcal{D}_S}(h) + L \cdot \text{MMD}(P_S, P_T) \tag{32}$$

This completes the proof.

Theorem C.5 provides the theoretical foundation for our Distribution Shift Detector. The bound reveals that the expected risk on the target distribution is controlled by two terms: the risk on the source distribution $\epsilon_{\mathcal{D}_S}(h)$, which reflects how well the current experts perform on historical data, and the distribution discrepancy $L \cdot \text{MMD}(P_S, P_T)$, which quantifies the shift between distributions.

When the MMD distance between the reference window and the current window increases, the upper bound on the target risk grows proportionally. This implies that experts trained on historical data may no longer guarantee low risk on the shifted distribution. The adaptive threshold $\epsilon = \mu_H + \lambda \cdot \sigma_H$ in our detector provides a practical criterion: when the observed MMD exceeds this threshold, it signals that the distribution has shifted significantly beyond historical fluctuations, and the current expert pool may be inadequate for the new data regime. This motivates the instantiation of new experts specifically adapted to the drifted distribution.

## D. Full Results

In this section, we provide the complete performance comparison of Dynamic TMoE against state-of-the-art baselines on all nine datasets. We compare our model with nine widely recognized SOTA baselines, including TFPS (Sun et al., 2025), ST-MTM (Seo & Lim, 2025), RAFT (Han et al., 2025), TimeMixer (Wang et al., 2024), FITS (Xu et al., 2024), PatchTST (Nie et al., 2023), TimesNet (Wu et al., 2023), DLinear (Zeng et al., 2023), and FEDformer (Zhou et al., 2022). Table 8 reports the comprehensive multivariate forecasting results.

As shown in Table 8, our proposed model consistently maintains a significant performance advantage in almost all prediction horizons. For instance, on the Weather and ETTm1/m2 datasets, our model achieved the best or second-best performance across all prediction horizons. A granular comparison with TFPS, the leading MoE-based baseline, reveals that Dynamic TMoE's advantage is systemic rather than situational. Our model outperforms TFPS in the vast majority of individual horizon settings across 8 out of 9 datasets. This sustained advantage strongly demonstrates that incorporating dynamic evolution during training is more effective than static clustering for handling complex non-stationary patterns.

## E. Additional Baselines

We have expanded our empirical evaluation to include a broader set of recent state-of-the-art (SOTA) baselines. This supplementary experiment aims to comprehensively validate the robustness and superiority of the proposed Dynamic TMoE framework. We incorporate six additional competitive models for comparison: MoE-based Models: MoLE (Ni et al., 2024) and FreqMoE (Liu, 2025). Transformer-based Models: iTransformer (Liu et al., 2024) and PerimidFormer (Wu et al., 2024). Other Advanced Architectures: TSLANet (Eldele et al., 2024) and TimeExpert (Ma et al., 2025).

As demonstrated in Table 9, Dynamic TMoE maintains a highly competitive edge against these recently proposed architectures. Specifically, our framework achieves the best performance on 5 out of the 9 datasets (Weather, ETTh1, ETTm1, ETTm2, and ILI).

## F. Full Ablations

In this section, we provide detailed definitions of the model variants used in our ablation studies and present the comprehensive results. This supplements the findings in Section 4.3 of the main paper.

### F.1. Ablation on Drift-Aware Adaptation and Temporal Memory Router

**Model Configurations:**

① **Dynamic TMoE:** Our proposed model integrating the drift-aware adaptation strategy, GRU-based temporal memory router and anomaly state memory.

② **w/o Drift-Aware Adaptation:** This variant removes the MMD drift detector and the dynamic expert expansion

mechanism. The model relies solely on the fixed base experts trained on the initial data, prohibiting the instantiation and targeted alignment of new experts.

③ **w/ Linear Router:** Replaces the GRU-based router with a simple linear projection layer. The routing decision depends only on the current patch features, ignoring historical routing information.

④ **w/ MLP Router:** Replaces the GRU-based router with a Multilayer Perceptron (MLP). While non-linear, it still lacks the explicit temporal recurrence to maintain routing consistency.

⑤ **w/o Anomaly Memory:** Removes the Anomaly State Repository. The router does not have access to stored historical drift states when making decisions.

Table 10 presents the full comparison results. It is evident that removing the dynamic mechanism significantly degrades performance on non-stationary datasets. Furthermore, both Linear and MLP routers underperform compared to the GRU router, highlighting the importance of temporal memory in stabilizing expert selection.

### F.2. Ablation on Expert Diversity and Relation Layer

**Model Configurations:**

① **Dynamic TMoE:** Our proposed model using a diverse set of experts: Identity, Trend, Seasonality and Fluctuation experts.

② **w/o Relation Layer:** The full model but with the Cyclic Relation Layer removed from all experts. This variant treats variables independently without modeling inter-variable correlations.

③ **All Identity:** Replaces all base and dynamic experts with Identity experts.

④ **All Trend:** Replaces all base and dynamic experts with Trend experts.

⑤ **All Seasonality:** Replaces all base and dynamic experts with Seasonality experts.

⑥ **All Fluctuation:** Replaces all base and dynamic experts with Fluctuation experts.

Table 11 shows the impact of expert diversity. The heterogeneous ensemble consistently outperforms any single-type homogeneous ensemble, proving that real-world time series contain mixed patterns that require specialized processing. The significant drop in Model ② further confirms that modeling cross-variable relationships via the Cyclic Relation Layer is crucial for multivariate forecasting.

### F.3. Ablation on Adaptation Strategies

**Model Configurations:**

① **Dynamic TMoE:** Our proposed model using MMD for drift detection, Post-Addition Alignment strategy, Drift Pattern Profiler for selection, and the expert pruning mechanism.

② **w/ Simple Detection:** Replaces the MMD detector with a statistical detector based on mean and variance shifts.

③ **Fine-tune Base Experts:** Instead of using our Post-Addition Alignment strategy, ③ directly fine-tunes the parameters of the existing base experts on drift data.

④ **w/ Random Selection:** Instead of using the Drift Pattern Profiler to determine the type of the new expert, a type is chosen randomly from the available pool.

⑤ **w/o Expert Pruning:** Disables the mechanism that removes low-usage experts. This expert pool is allowed to grow indefinitely.

Table 12 details the effectiveness of each component in the adaptation strategies. The MMD detector (Model ①) shows superior sensitivity to complex distribution shifts compared to simple statistics (Model ②). Notably, Model ③ suffers from catastrophic forgetting, leading to poor performance on datasets with recurrent patterns. The comparison between Model ① and Model ④ confirms that our Drift Pattern Profiler effectively identifies the dominant pattern in the drift data. The results of Model ⑤ prove the effectiveness of expert pruning.

## G. Showcases

To qualitatively evaluate the forecasting performance, we visualize the prediction results from the test sets of representative benchmarks, including Electricity, Traffic, and ILI (Figure 5, 6, 7). These figures compare the forecasting curves of Dynamic TMoE against competitive state-of-the-art baselines, demonstrating that Dynamic TMoE exhibits superior performance.

## H. Disentangling Architectural Innovation from Raw Capacity Scaling

In this section, we provide a detailed clarification to disentangle the performance improvements of Dynamic TMoE from mere model capacity scaling. While conventional MoE architectures often rely on increasing the number of experts to boost representational capacity, our core innovation lies in the unified **"Perception-Decision-Adaptation"** framework. This framework shifts the MoE paradigm from a static, memoryless architecture into an evolvable, history-aware dynamic system. The empirical evidence across our ablation and sensitivity studies demonstrates that our evolutionary logic, rather than raw parameter count, drives the state-of-the-art performance.

### H.1. Performance Degradation with Excess Capacity

To verify that the performance gains are not simply a byproduct of adding more experts, we analyze the model's sensitivity to the drift expert pool size. As illustrated in Figure 3(c), increasing the model's capacity by enlarging the drift expert pool size actually degrades forecasting performance. This empirical degradation indicates that larger expert pools introduce unnecessary complexity and overfitting risks. Dynamic TMoE performs optimally with a compact, dynamically managed pool, proving that our success relies on precise expert utilization rather than capacity maximization.

### H.2. The Necessity of the Dynamic Evolution Mechanism

The critical role of our evolutionary logic over static capacity is further validated by our ablation studies. As demonstrated in Table 2, disabling the Drift-Aware Adaptation mechanism within the base architecture leads to a direct performance drop. This confirms that static architectures, even those equipped with multiple experts, fail to accommodate evolving distributions effectively without our dynamic tracking and adaptation mechanisms.

### H.3. System-Level Synergy over Individual Modules

Dynamic TMoE's superiority is derived from the coupling of its architectural innovations rather than the stacking of independent modules. Our Temporal Memory Router fundamentally rethinks MoE routing by formulating it as a sequential modeling task, ensuring context-aware expert selection and mitigating the erratic switching inherent to memoryless designs. Concurrently, our Heterogeneous Experts serve as a dynamic reservoir of distinct inductive biases, effectively disentangling complex temporal dynamics in a way that homogeneous expert designs cannot (as validated in Table 3). Together, these components ensure that the model autonomously shifts its focus and adapts to temporal distribution shifts without relying on brute-force capacity scaling.

## I. Limitations and Future Work

### I.1. Limitations

**Computational Complexity of Drift Detection.** The primary computational overhead in our framework resides in the Maximum Mean Discrepancy (MMD) drift detector. The kernel matrix calculation entails a quadratic complexity $O(N^2)$ relative to the window size $N$, which presents a challenge for high-frequency or long-horizon forecasting. To ensure scalability during the training phase, we currently employ a sampling-based approximation to estimate MMD scores. While this pragmatic trade-off substantially reduces per-iteration latency, it may marginally decrease sensitivity to subtle distribution shifts compared to an exact calculation.

**Sensitivity to Hyperparameter Configuration.** The efficacy of our dynamic framework depends on the coordinated tuning of a broad set of hyperparameters across the perception, decision and adaptation modules. Beyond the drift detector's settings, the model's behavior is also governed by expert management constraints, such as the maximum expert capacity, pruning frequency and low-usage removal thresholds. Currently, these parameters are selected empirically based on validation performance. The interplay between these hyperparameters creates a complex search space, where suboptimal configurations

can lead to instability. For instance, an overly aggressive pruning threshold might prematurely discard useful experts, while a loose capacity limit could dilute model specialization. The lack of an automated, adaptive hyperparameter optimization mechanism remains a barrier to effortless deployment on new datasets.

**Cold-Start for New Experts.** Dynamic TMoE introduces a transient cold-start phase for initialization of new experts. To ensure that newly added experts possess relevant prior knowledge, they are initialized from a base template rather than random weights. This design represents a principled trade-off between immediate responsiveness and long-term specialization. Currently, the expert will have a brief warm-up period for adapting to the new distribution. During this phase, the model may have a temporary variance in prediction error until the new parameters stabilize.

## I.2. Future Work

**Online Learning and Test-Time Adaptation.** While Dynamic TMoE ensures inference stability and computational efficiency by maintaining a fixed architecture post-deployment, this design represents a deliberate trade-off between robustness and real-time plasticity. Currently, architectural evolution is restricted to the training phase to avoid the risks of catastrophic forgetting or instability inherent in streaming scenarios. A promising future direction is to extend our framework into an online learning setting, allowing the Evolvable Expert Manager to continuously instantiate or update experts during inference. This would enable the model to handle indefinite horizons and extreme distribution shifts without the need for periodic offline re-training.

**Parameter-Efficient Adaptation.** To address the computational overhead of adding full experts, future work will explore Parameter-Efficient Fine-Tuning (PEFT) techniques, such as Low-Rank Adaptation (LoRA) or Adapters. Instead of instantiating distinct neural networks for each new regime, we could maintain a single frozen backbone and dynamically inject lightweight, trainable rank-decomposition matrices. This would significantly reduce the memory footprint while maintaining the plasticity required for drift adaptation.

**Integration with Time Series Foundation Models.** While Dynamic TMoE is presented here as a specialized architecture, its core mechanisms, including structural plasticity and state-aware routing, are fundamentally modular. Rather than treating it as a standalone model, we view it as a scalable strategy for enhancing Time Series Foundation Models. Current foundation models often struggle with zero-shot generalization when encountering out-of-distribution shifts after pre-training. By integrating our evolvable expert pool and temporal memory routing, these large-scale models could theoretically achieve dynamic adaptation to downstream tasks. This integration would provide a mechanism to accommodate novel data regimes via lightweight expert instantiation, preserving the foundational knowledge while extending the applicability of models to unseen distributions without the overhead of full-model retraining.

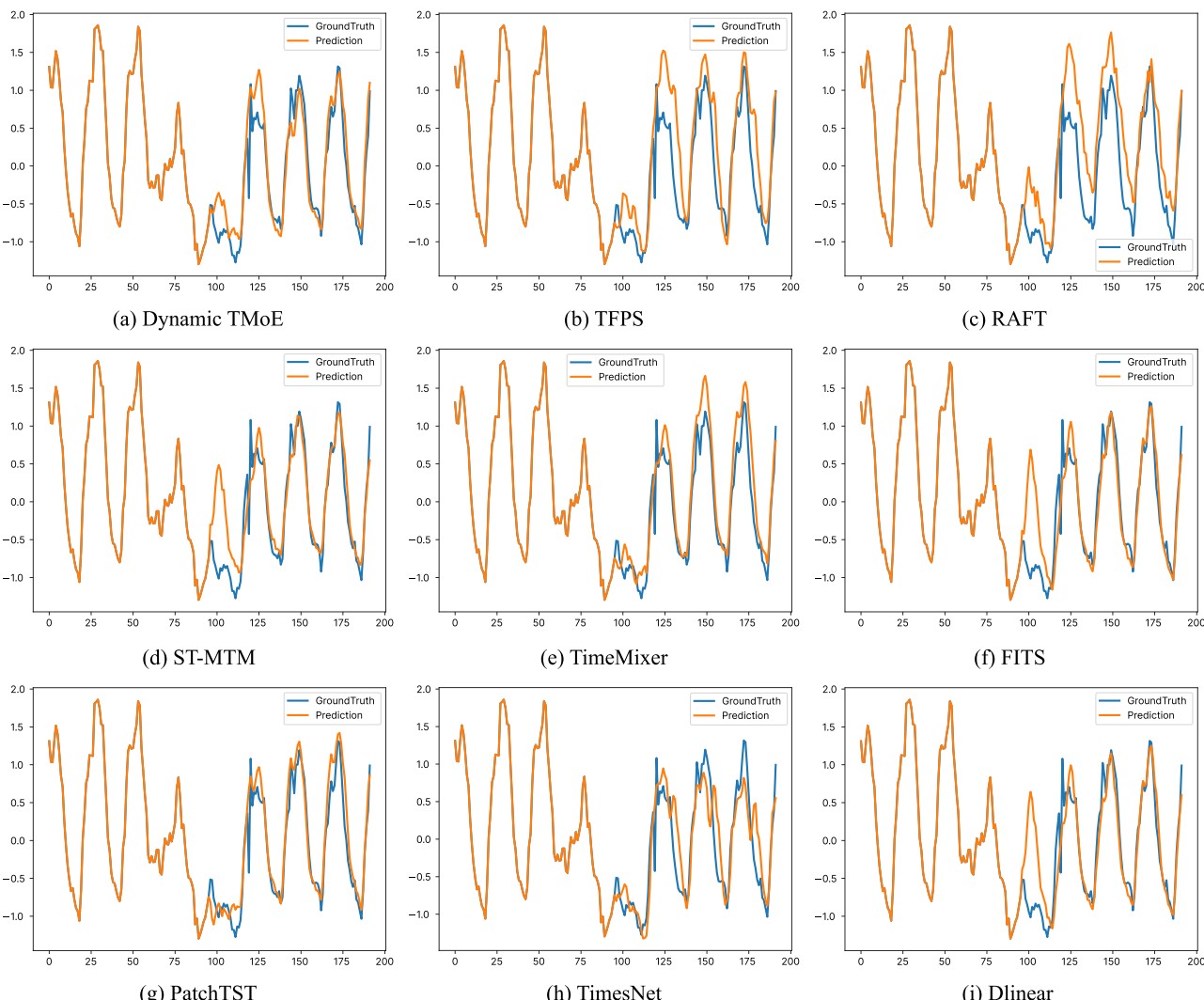

*Figure 5.* Visualization of forecasting results on the Electricity dataset with an input-96-predict-96 setting. The blue lines represent the ground truth, while the orange lines indicate the model predictions.

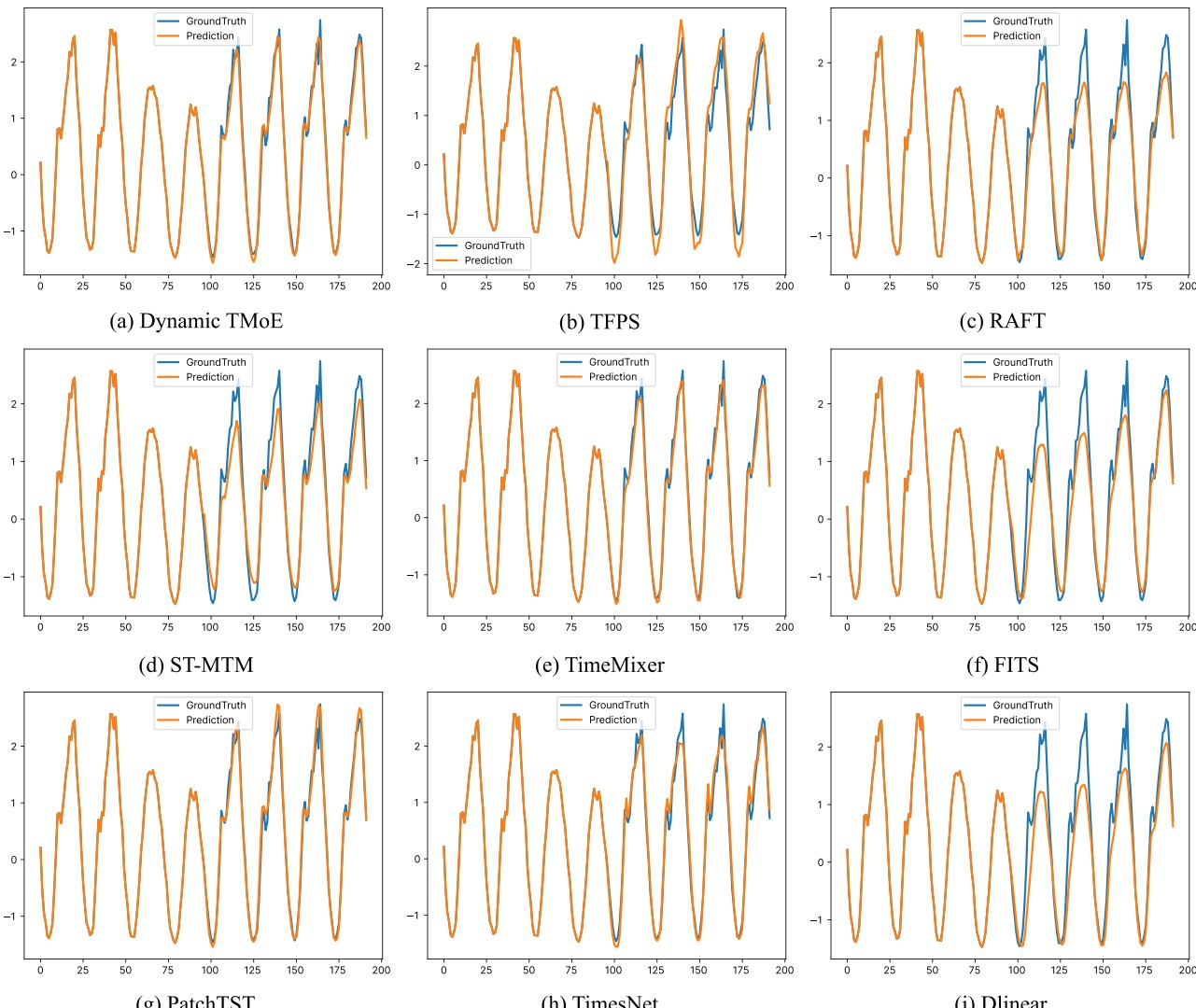

*Figure 6.* Visualization of forecasting results on the Traffic dataset with an input-96-predict-96 setting. The blue lines represent the ground truth, while the orange lines indicate the model predictions.

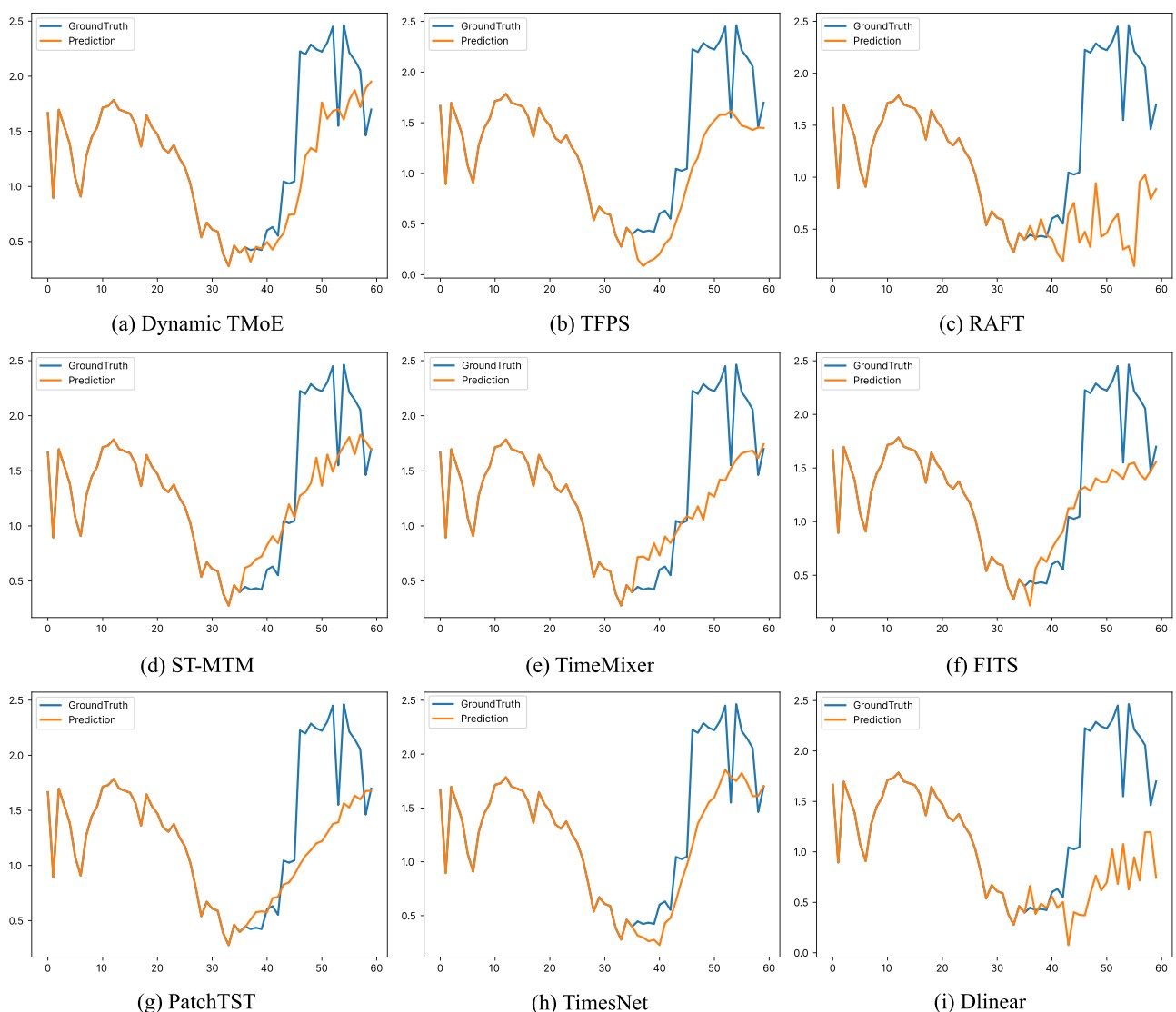

*Figure 7.* Visualization of forecasting results on the ILI dataset with an input-36-predict-24 setting. The blue lines represent the ground truth, while the orange lines indicate the model predictions.

*Table 7.* Detailed hyperparameter settings for each dataset and prediction horizon ($T$). The parameters include Patch Length ($P$), Stride ($S$), Initial Learning Rate (LR), Dropout, number of MoE layers, number of RNN router layers, and Batch Size.

| Dataset | Horizon ($T$) | Patching | | Training | | Architecture | | Batch Size |
|---|---|---|---|---|---|---|---|---|
| | | $P$ | $S$ | LR | Dropout | MoE Layers | Router RNN Layers | |
| ETTh1 | 96 | 48 | 12 | $1.2 \times 10^{-3}$ | 0.4 | 2 | 1 | 256 |
| | 192 | 48 | 12 | $8 \times 10^{-4}$ | 0.6 | 2 | 2 | 256 |
| | 336 | 48 | 12 | $4 \times 10^{-4}$ | 0.3 | 1 | 2 | 256 |
| | 720 | 48 | 12 | $4 \times 10^{-4}$ | 0.8 | 1 | 1 | 256 |
| ETTh2 | 96 | 24 | 6 | $8 \times 10^{-4}$ | 0.8 | 2 | 1 | 256 |
| | 192 | 24 | 6 | $4 \times 10^{-4}$ | 0.6 | 2 | 1 | 256 |
| | 336 | 24 | 6 | $4 \times 10^{-4}$ | 0.6 | 2 | 1 | 256 |
| | 720 | 24 | 6 | $4 \times 10^{-4}$ | 0.8 | 2 | 1 | 256 |
| ETTm1 | 96 | 48 | 12 | $4 \times 10^{-4}$ | 0.4 | 2 | 2 | 256 |
| | 192 | 48 | 12 | $4 \times 10^{-4}$ | 0.4 | 2 | 2 | 256 |
| | 336 | 24 | 12 | $1.6 \times 10^{-3}$ | 0.6 | 2 | 2 | 256 |
| | 720 | 48 | 12 | $4 \times 10^{-4}$ | 0.4 | 2 | 2 | 256 |
| ETTm2 | 96 | 48 | 6 | $8 \times 10^{-4}$ | 0.8 | 2 | 1 | 256 |
| | 192 | 48 | 6 | $8 \times 10^{-4}$ | 0.8 | 1 | 2 | 256 |
| | 336 | 48 | 6 | $4 \times 10^{-4}$ | 0.8 | 1 | 1 | 256 |
| | 720 | 24 | 12 | $1.2 \times 10^{-3}$ | 0.6 | 1 | 2 | 256 |
| Traffic | 96 | 48 | 6 | $2.4 \times 10^{-3}$ | 0.1 | 2 | 2 | 32 |
| | 192 | 48 | 6 | $1.2 \times 10^{-3}$ | 0.1 | 2 | 2 | 32 |
| | 336 | 48 | 6 | $1.8 \times 10^{-3}$ | 0.1 | 2 | 1 | 32 |
| | 720 | 48 | 6 | $2.4 \times 10^{-3}$ | 0.1 | 2 | 1 | 32 |
| Electricity | 96 | 24 | 6 | $8 \times 10^{-4}$ | 0.1 | 2 | 2 | 32 |
| | 192 | 24 | 6 | $8 \times 10^{-4}$ | 0.3 | 2 | 2 | 32 |
| | 336 | 24 | 6 | $1.2 \times 10^{-3}$ | 0.3 | 2 | 1 | 32 |
| | 720 | 24 | 6 | $8 \times 10^{-4}$ | 0.3 | 2 | 2 | 32 |
| Weather | 96 | 96 | 24 | $1.2 \times 10^{-3}$ | 0.1 | 1 | 2 | 256 |
| | 192 | 96 | 24 | $4 \times 10^{-4}$ | 0.1 | 1 | 1 | 256 |
| | 336 | 96 | 48 | $6 \times 10^{-4}$ | 0.1 | 2 | 1 | 256 |
| | 720 | 48 | 48 | $1.2 \times 10^{-3}$ | 0.1 | 1 | 1 | 256 |
| ILI | 24 | 24 | 4 | $8 \times 10^{-4}$ | 0.1 | 2 | 1 | 64 |
| | 36 | 24 | 2 | $4 \times 10^{-4}$ | 0.1 | 2 | 1 | 64 |
| | 48 | 24 | 4 | $8 \times 10^{-4}$ | 0.3 | 2 | 1 | 64 |
| | 60 | 24 | 2 | $4 \times 10^{-4}$ | 0.3 | 1 | 1 | 64 |
| Exchange | 96 | 16 | 8 | $4 \times 10^{-4}$ | 0.3 | 2 | 1 | 128 |
| | 192 | 16 | 8 | $4 \times 10^{-4}$ | 0.3 | 1 | 2 | 128 |
| | 336 | 32 | 8 | $8 \times 10^{-4}$ | 0.3 | 1 | 2 | 128 |
| | 720 | 16 | 8 | $4 \times 10^{-4}$ | 0.5 | 2 | 2 | 128 |

*Table 8.* Full multivariate time series forecasting results. This table reports the detailed MSE and MAE values for each specific prediction horizon. The horizons are set to $\{24, 36, 48, 60\}$ for the ILI dataset and $\{96, 192, 336, 720\}$ for the others. The look-back window length is fixed at 36 for ILI and 96 for the remaining datasets. The best results are in **bold** and the second best are underlined.

| Models | Horizon | Dynamic TMoE (Ours) | | TFPS (2025) | | ST-MTM (2025) | | RAFT (2025) | | TimeMixer (2024) | | FITS (2024) | | PatchTST (2023) | | TimesNet (2023) | | DLinear (2023) | | FEDformer (2022) | |
|---|---|---|---|---|---|---|---|---|---|---|---|---|---|---|---|---|---|---|---|---|---|
| | | MSE | MAE | MSE | MAE | MSE | MAE | MSE | MAE | MSE | MAE | MSE | MAE | MSE | MAE | MSE | MAE | MSE | MAE | MSE | MAE |
| Weather | 96 | 0.154 | 0.201 | 0.154 | 0.202 | 0.181 | 0.229 | 0.189 | 0.246 | 0.163 | 0.209 | 0.165 | 0.212 | 0.178 | 0.219 | 0.172 | 0.220 | 0.197 | 0.255 | 0.217 | 0.296 |
| | 192 | 0.202 | 0.246 | 0.205 | 0.249 | 0.227 | 0.270 | 0.239 | 0.289 | 0.208 | 0.250 | 0.214 | 0.256 | 0.224 | 0.259 | 0.219 | 0.261 | 0.239 | 0.297 | 0.276 | 0.336 |
| | 336 | 0.260 | 0.289 | 0.262 | 0.289 | 0.281 | 0.310 | 0.291 | 0.329 | 0.251 | 0.287 | 0.267 | 0.293 | 0.278 | 0.298 | 0.280 | 0.306 | 0.283 | 0.332 | 0.339 | 0.380 |
| | 720 | 0.343 | 0.344 | 0.344 | 0.342 | 0.359 | 0.364 | 0.367 | 0.380 | 0.339 | 0.341 | 0.348 | 0.345 | 0.350 | 0.346 | 0.365 | 0.359 | 0.348 | 0.385 | 0.403 | 0.428 |
| | Avg | 0.240 | 0.270 | 0.241 | 0.271 | 0.262 | 0.293 | 0.271 | 0.311 | 0.240 | 0.272 | 0.249 | 0.277 | 0.258 | 0.281 | 0.259 | 0.287 | 0.267 | 0.317 | 0.309 | 0.360 |
| Exchange | 96 | 0.080 | 0.197 | 0.083 | 0.205 | 0.099 | 0.222 | 0.084 | 0.200 | 0.092 | 0.211 | 0.084 | 0.204 | 0.084 | 0.200 | 0.107 | 0.234 | 0.088 | 0.216 | 0.148 | 0.278 |
| | 192 | 0.170 | 0.293 | 0.174 | 0.297 | 0.195 | 0.315 | 0.177 | 0.296 | 0.195 | 0.312 | 0.177 | 0.301 | 0.178 | 0.299 | 0.226 | 0.344 | 0.173 | 0.305 | 0.271 | 0.380 |
| | 336 | 0.325 | 0.412 | 0.310 | 0.398 | 0.354 | 0.433 | 0.353 | 0.423 | 0.346 | 0.424 | 0.320 | 0.409 | 0.365 | 0.431 | 0.367 | 0.448 | 0.307 | 0.419 | 0.460 | 0.500 |
| | 720 | 0.828 | 0.686 | 1.011 | 0.756 | 0.983 | 0.756 | 1.014 | 0.753 | 0.862 | 0.695 | 0.832 | 0.688 | 0.914 | 0.714 | 0.964 | 0.746 | 0.782 | 0.669 | 1.195 | 0.841 |
| | Avg | 0.351 | 0.397 | 0.395 | 0.414 | 0.408 | 0.432 | 0.407 | 0.418 | 0.374 | 0.411 | 0.353 | 0.400 | 0.385 | 0.411 | 0.416 | 0.443 | 0.337 | 0.402 | 0.519 | 0.500 |
| ETTh1 | 96 | 0.369 | 0.392 | 0.398 | 0.413 | 0.376 | 0.396 | 0.387 | 0.414 | 0.375 | 0.400 | 0.385 | 0.393 | 0.393 | 0.408 | 0.384 | 0.402 | 0.383 | 0.396 | 0.376 | 0.419 |
| | 192 | 0.417 | 0.420 | 0.423 | 0.423 | 0.422 | 0.423 | 0.423 | 0.431 | 0.429 | 0.421 | 0.435 | 0.422 | 0.445 | 0.434 | 0.436 | 0.429 | 0.433 | 0.426 | 0.420 | 0.448 |
| | 336 | 0.459 | 0.445 | 0.484 | 0.461 | 0.458 | 0.443 | 0.456 | 0.445 | 0.484 | 0.458 | 0.475 | 0.444 | 0.484 | 0.451 | 0.491 | 0.469 | 0.491 | 0.467 | 0.459 | 0.465 |
| | 720 | 0.471 | 0.463 | 0.488 | 0.476 | 0.472 | 0.469 | 0.463 | 0.464 | 0.498 | 0.482 | 0.463 | 0.459 | 0.480 | 0.471 | 0.521 | 0.500 | 0.528 | 0.519 | 0.506 | 0.507 |
| | Avg | 0.429 | 0.430 | 0.448 | 0.443 | 0.432 | 0.433 | 0.432 | 0.438 | 0.447 | 0.440 | 0.439 | 0.429 | 0.451 | 0.441 | 0.458 | 0.450 | 0.459 | 0.452 | 0.440 | 0.460 |
| ETTh2 | 96 | 0.283 | 0.335 | 0.313 | 0.355 | 0.274 | 0.333 | 0.296 | 0.350 | 0.289 | 0.341 | 0.290 | 0.339 | 0.294 | 0.343 | 0.340 | 0.374 | 0.329 | 0.380 | 0.346 | 0.388 |
| | 192 | 0.358 | 0.388 | 0.405 | 0.410 | 0.359 | 0.384 | 0.384 | 0.403 | 0.372 | 0.392 | 0.377 | 0.391 | 0.377 | 0.393 | 0.402 | 0.414 | 0.431 | 0.443 | 0.429 | 0.439 |
| | 336 | 0.408 | 0.427 | 0.392 | 0.415 | 0.394 | 0.415 | 0.426 | 0.442 | 0.386 | 0.414 | 0.416 | 0.425 | 0.381 | 0.409 | 0.452 | 0.452 | 0.459 | 0.462 | 0.496 | 0.487 |
| | 720 | 0.422 | 0.442 | 0.410 | 0.433 | 0.407 | 0.433 | 0.434 | 0.460 | 0.412 | 0.434 | 0.417 | 0.436 | 0.412 | 0.433 | 0.462 | 0.468 | 0.774 | 0.631 | 0.463 | 0.474 |
| | Avg | 0.368 | 0.398 | 0.380 | 0.403 | 0.359 | 0.391 | 0.385 | 0.413 | 0.365 | 0.395 | 0.375 | 0.398 | 0.366 | 0.395 | 0.414 | 0.427 | 0.498 | 0.479 | 0.434 | 0.447 |
| ETTm1 | 96 | 0.311 | 0.352 | 0.327 | 0.367 | 0.340 | 0.373 | 0.329 | 0.371 | 0.320 | 0.357 | 0.354 | 0.375 | 0.321 | 0.360 | 0.338 | 0.375 | 0.346 | 0.374 | 0.379 | 0.419 |
| | 192 | 0.358 | 0.381 | 0.374 | 0.395 | 0.382 | 0.393 | 0.363 | 0.388 | 0.361 | 0.381 | 0.392 | 0.393 | 0.362 | 0.384 | 0.374 | 0.387 | 0.383 | 0.393 | 0.426 | 0.441 |
| | 336 | 0.388 | 0.402 | 0.401 | 0.408 | 0.413 | 0.413 | 0.391 | 0.408 | 0.390 | 0.404 | 0.425 | 0.414 | 0.392 | 0.402 | 0.410 | 0.411 | 0.417 | 0.418 | 0.445 | 0.459 |
| | 720 | 0.447 | 0.441 | 0.479 | 0.456 | 0.469 | 0.444 | 0.444 | 0.438 | 0.454 | 0.441 | 0.486 | 0.448 | 0.450 | 0.435 | 0.478 | 0.450 | 0.479 | 0.457 | 0.543 | 0.490 |
| | Avg | 0.376 | 0.394 | 0.395 | 0.407 | 0.401 | 0.406 | 0.382 | 0.401 | 0.381 | 0.396 | 0.414 | 0.408 | 0.381 | 0.395 | 0.400 | 0.406 | 0.406 | 0.410 | 0.448 | 0.452 |
| ETTm2 | 96 | 0.168 | 0.253 | 0.170 | 0.255 | 0.177 | 0.260 | 0.177 | 0.266 | 0.172 | 0.258 | 0.183 | 0.266 | 0.178 | 0.260 | 0.187 | 0.267 | 0.187 | 0.281 | 0.203 | 0.287 |
| | 192 | 0.232 | 0.296 | 0.235 | 0.296 | 0.242 | 0.302 | 0.243 | 0.308 | 0.237 | 0.299 | 0.247 | 0.305 | 0.249 | 0.307 | 0.249 | 0.309 | 0.272 | 0.349 | 0.269 | 0.328 |
| | 336 | 0.290 | 0.334 | 0.297 | 0.335 | 0.302 | 0.340 | 0.302 | 0.346 | 0.298 | 0.340 | 0.307 | 0.342 | 0.313 | 0.346 | 0.321 | 0.351 | 0.343 | 0.395 | 0.325 | 0.366 |
| | 720 | 0.388 | 0.391 | 0.401 | 0.397 | 0.397 | 0.396 | 0.402 | 0.404 | 0.391 | 0.396 | 0.407 | 0.398 | 0.400 | 0.398 | 0.408 | 0.403 | 0.439 | 0.444 | 0.421 | 0.415 |
| | Avg | 0.269 | 0.318 | 0.276 | 0.321 | 0.280 | 0.325 | 0.281 | 0.331 | 0.275 | 0.323 | 0.286 | 0.328 | 0.285 | 0.328 | 0.291 | 0.333 | 0.310 | 0.367 | 0.305 | 0.349 |
| Traffic | 96 | 0.459 | 0.280 | 0.427 | 0.296 | 0.540 | 0.331 | 0.566 | 0.373 | 0.462 | 0.285 | 0.650 | 0.390 | 0.477 | 0.305 | 0.593 | 0.321 | 0.650 | 0.397 | 0.587 | 0.366 |
| | 192 | 0.470 | 0.280 | 0.445 | 0.298 | 0.540 | 0.332 | 0.526 | 0.344 | 0.473 | 0.296 | 0.601 | 0.364 | 0.471 | 0.299 | 0.617 | 0.336 | 0.600 | 0.372 | 0.604 | 0.373 |
| | 336 | 0.476 | 0.293 | 0.459 | 0.307 | 0.552 | 0.335 | 0.519 | 0.334 | 0.498 | 0.296 | 0.609 | 0.366 | 0.485 | 0.305 | 0.629 | 0.336 | 0.606 | 0.374 | 0.621 | 0.383 |
| | 720 | 0.512 | 0.301 | 0.496 | 0.313 | 0.590 | 0.351 | 0.535 | 0.336 | 0.506 | 0.313 | 0.647 | 0.387 | 0.518 | 0.325 | 0.640 | 0.350 | 0.646 | 0.396 | 0.626 | 0.382 |
| | Avg | 0.479 | 0.288 | 0.457 | 0.304 | 0.556 | 0.337 | 0.537 | 0.347 | 0.485 | 0.298 | 0.627 | 0.377 | 0.488 | 0.309 | 0.620 | 0.336 | 0.625 | 0.385 | 0.610 | 0.376 |
| Electricity | 96 | 0.145 | 0.243 | 0.149 | 0.236 | 0.188 | 0.284 | 0.174 | 0.283 | 0.153 | 0.247 | 0.199 | 0.276 | 0.174 | 0.259 | 0.168 | 0.272 | 0.195 | 0.277 | 0.193 | 0.308 |
| | 192 | 0.163 | 0.262 | 0.162 | 0.253 | 0.192 | 0.288 | 0.170 | 0.275 | 0.166 | 0.256 | 0.198 | 0.277 | 0.178 | 0.265 | 0.184 | 0.289 | 0.194 | 0.280 | 0.201 | 0.315 |
| | 336 | 0.174 | 0.272 | 0.200 | 0.310 | 0.207 | 0.303 | 0.179 | 0.283 | 0.185 | 0.277 | 0.213 | 0.293 | 0.196 | 0.282 | 0.198 | 0.300 | 0.207 | 0.296 | 0.214 | 0.329 |
| | 720 | 0.198 | 0.296 | 0.220 | 0.320 | 0.246 | 0.333 | 0.212 | 0.308 | 0.225 | 0.310 | 0.254 | 0.325 | 0.237 | 0.316 | 0.220 | 0.320 | 0.243 | 0.328 | 0.246 | 0.355 |
| | Avg | 0.170 | 0.268 | 0.183 | 0.280 | 0.208 | 0.302 | 0.184 | 0.287 | 0.182 | 0.273 | 0.216 | 0.293 | 0.196 | 0.281 | 0.193 | 0.295 | 0.210 | 0.296 | 0.214 | 0.327 |
| ILI | 24 | 1.930 | 0.873 | 3.059 | 1.069 | 2.764 | 1.064 | 5.589 | 1.583 | 2.918 | 1.153 | 2.385 | 1.050 | 1.758 | 0.827 | 2.317 | 0.934 | 3.052 | 1.245 | 3.228 | 1.260 |
| | 36 | 2.143 | 0.912 | 2.513 | 0.963 | 2.783 | 1.065 | 6.365 | 1.739 | 2.730 | 1.081 | 2.417 | 1.069 | 1.577 | 0.809 | 1.972 | 0.920 | 2.804 | 1.153 | 2.679 | 1.080 |
| | 48 | 1.914 | 0.870 | 2.395 | 0.941 | 2.844 | 1.083 | 6.226 | 1.719 | 2.730 | 1.083 | 2.641 | 1.137 | 1.978 | 0.849 | 2.238 | 0.940 | 2.829 | 1.161 | 2.622 | 1.078 |
| | 60 | 1.939 | 0.895 | 2.599 | 0.992 | 2.890 | 1.092 | 5.486 | 1.590 | 3.386 | 1.265 | 2.818 | 1.173 | 1.627 | 0.808 | 2.027 | 0.928 | 2.973 | 1.193 | 2.857 | 1.157 |
| | Avg | 1.981 | 0.888 | 2.642 | 0.991 | 2.820 | 1.076 | 5.916 | 1.658 | 2.941 | 1.145 | 2.565 | 1.107 | 1.735 | 0.823 | 2.139 | 0.931 | 2.915 | 1.188 | 2.847 | 1.144 |

*Table 9.* Multivariate time series forecasting results compared with additional baselines. The best results are in **bold** and the second best are underlined.

| Models | Dynamic TMoE (Ours) | | MoLE (Baseline) | | FreqMoE (Baseline) | | iTransformer (Baseline) | | PerimidFormer (Baseline) | | TSLANet (Baseline) | | TimeExpert (Baseline) | |
|---|---|---|---|---|---|---|---|---|---|---|---|---|---|---|
| Metric | MSE | MAE | MSE | MAE | MSE | MAE | MSE | MAE | MSE | MAE | MSE | MAE | MSE | MAE |
| Weather | **0.240** | **0.270** | 0.283 | 0.331 | 0.263 | 0.286 | 0.258 | 0.278 | 0.257 | 0.281 | 0.259 | 0.282 | 0.251 | 0.277 |
| Exchange | 0.351 | 0.397 | **0.286** | **0.378** | 0.370 | 0.408 | 0.360 | 0.403 | 0.361 | 0.402 | 0.361 | 0.404 | 0.352 | 0.398 |
| ETTh1 | **0.429** | **0.430** | 0.475 | 0.464 | 0.458 | 0.441 | 0.454 | 0.448 | 0.460 | 0.449 | 0.458 | 0.454 | 0.430 | 0.435 |
| ETTh2 | 0.368 | 0.398 | 0.732 | 0.558 | **0.364** | **0.395** | 0.383 | 0.407 | 0.389 | 0.408 | 0.368 | 0.399 | 0.374 | 0.404 |
| ETTm1 | **0.376** | **0.394** | 0.418 | 0.420 | 0.387 | 0.399 | 0.407 | 0.410 | 0.398 | 0.409 | 0.388 | 0.401 | 0.389 | 0.402 |
| ETTm2 | **0.269** | **0.318** | 0.406 | 0.403 | 0.279 | 0.323 | 0.288 | 0.332 | 0.290 | 0.330 | 0.283 | 0.327 | 0.283 | 0.329 |
| Traffic | 0.479 | 0.288 | 0.528 | 0.347 | 0.522 | 0.339 | **0.428** | **0.282** | 0.461 | 0.304 | 0.494 | 0.314 | 0.500 | 0.322 |
| Electricity | 0.170 | 0.268 | 0.191 | 0.284 | 0.205 | 0.289 | 0.178 | 0.270 | **0.169** | **0.262** | 0.187 | 0.278 | 0.203 | 0.294 |
| ILI | **1.981** | **0.888** | 2.630 | 1.044 | 4.083 | 1.472 | 2.257 | 0.967 | 2.210 | 0.914 | 2.209 | 0.938 | 2.471 | 1.014 |

*Table 10.* Full results of the ablation study on dynamic mechanism, temporal memory router, and anomaly memory components. We compare the proposed Dynamic TMoE (Model ①) against variants that remove the dynamic mechanism (Model ②), replace the GRU router with Linear (Model ③) or MLP (Model ④) counterparts, and exclude the anomaly state memory (Model ⑤). *Note: For ILI dataset, the prediction lengths are 24, 36, 48, and 60, corresponding to the shown 96, 192, 336, and 720 positions respectively.*

| Model | Dynamic Mechanism | Temporal Memory Router | | | Anomaly Memory | Len. | ETTh1 | | ETTh2 | | ETTm1 | | ETTm2 | | Weather | | ILI | |
|---|---|---|---|---|---|---|---|---|---|---|---|---|---|---|---|---|---|---|
| | | GRU | Linear | MLP | | | MSE | MAE | MSE | MAE | MSE | MAE | MSE | MAE | MSE | MAE | MSE | MAE |
| ① | ✓ | ✓ | | | ✓ | 96 | 0.369 | 0.393 | 0.283 | 0.335 | 0.311 | 0.352 | 0.168 | 0.253 | 0.154 | 0.201 | 1.930 | 0.873 |
| | | | | | | 192 | 0.417 | 0.420 | 0.358 | 0.388 | 0.358 | 0.381 | 0.232 | 0.296 | 0.202 | 0.246 | 2.143 | 0.912 |
| | | | | | | 336 | 0.459 | 0.445 | 0.408 | 0.427 | 0.388 | 0.403 | 0.290 | 0.334 | 0.260 | 0.289 | 1.914 | 0.870 |
| | | | | | | 720 | 0.471 | 0.463 | 0.422 | 0.442 | 0.447 | 0.441 | 0.388 | 0.391 | 0.343 | 0.344 | 1.939 | 0.895 |
| | | | | | | Avg | **0.429** | **0.430** | **0.368** | **0.398** | **0.376** | **0.394** | **0.269** | **0.318** | **0.240** | **0.270** | **1.981** | **0.888** |
| ② | × | ✓ | | | ✓ | 96 | 0.377 | 0.394 | 0.285 | 0.337 | 0.324 | 0.361 | 0.175 | 0.259 | 0.163 | 0.209 | 1.889 | 0.828 |
| | | | | | | 192 | 0.416 | 0.419 | 0.364 | 0.394 | 0.360 | 0.382 | 0.237 | 0.300 | 0.208 | 0.249 | 2.151 | 0.917 |
| | | | | | | 336 | 0.469 | 0.442 | 0.413 | 0.430 | 0.393 | 0.406 | 0.295 | 0.337 | 0.268 | 0.293 | 2.282 | 0.922 |
| | | | | | | 720 | 0.485 | 0.471 | 0.422 | 0.442 | 0.452 | 0.442 | 0.390 | 0.395 | 0.346 | 0.344 | 2.262 | 0.979 |
| | | | | | | Avg | 0.436 | 0.432 | 0.371 | 0.401 | 0.382 | 0.397 | 0.274 | 0.323 | 0.246 | 0.274 | 2.146 | 0.911 |
| ③ | ✓ | | ✓ | | ✓ | 96 | 0.372 | 0.393 | 0.293 | 0.343 | 0.326 | 0.364 | 0.169 | 0.254 | 0.163 | 0.209 | 2.260 | 0.928 |
| | | | | | | 192 | 0.422 | 0.419 | 0.360 | 0.390 | 0.359 | 0.380 | 0.233 | 0.297 | 0.210 | 0.252 | 2.228 | 0.913 |
| | | | | | | 336 | 0.469 | 0.442 | 0.414 | 0.428 | 0.391 | 0.406 | 0.293 | 0.335 | 0.267 | 0.293 | 2.399 | 0.969 |
| | | | | | | 720 | 0.479 | 0.466 | 0.430 | 0.447 | 0.448 | 0.439 | 0.389 | 0.393 | 0.349 | 0.347 | 2.267 | 0.982 |
| | | | | | | Avg | 0.436 | 0.430 | 0.374 | 0.402 | 0.381 | 0.397 | 0.271 | 0.320 | 0.247 | 0.275 | 2.289 | 0.948 |
| ④ | ✓ | | | ✓ | ✓ | 96 | 0.373 | 0.393 | 0.288 | 0.338 | 0.323 | 0.360 | 0.172 | 0.256 | 0.157 | 0.203 | 2.352 | 0.943 |
| | | | | | | 192 | 0.421 | 0.421 | 0.373 | 0.396 | 0.359 | 0.380 | 0.234 | 0.298 | 0.207 | 0.249 | 2.446 | 0.931 |
| | | | | | | 336 | 0.474 | 0.446 | 0.411 | 0.429 | 0.391 | 0.404 | 0.296 | 0.338 | 0.267 | 0.293 | 2.136 | 0.885 |
| | | | | | | 720 | 0.485 | 0.475 | 0.437 | 0.451 | 0.458 | 0.444 | 0.399 | 0.398 | 0.347 | 0.346 | 2.025 | 0.909 |
| | | | | | | Avg | 0.438 | 0.434 | 0.377 | 0.404 | 0.383 | 0.397 | 0.275 | 0.322 | 0.245 | 0.273 | 2.240 | 0.917 |
| ⑤ | ✓ | ✓ | | | × | 96 | 0.369 | 0.393 | 0.288 | 0.339 | 0.325 | 0.363 | 0.169 | 0.253 | 0.159 | 0.205 | 2.469 | 0.962 |
| | | | | | | 192 | 0.417 | 0.421 | 0.368 | 0.395 | 0.363 | 0.383 | 0.235 | 0.298 | 0.209 | 0.251 | 2.546 | 0.968 |
| | | | | | | 336 | 0.477 | 0.445 | 0.412 | 0.428 | 0.393 | 0.406 | 0.295 | 0.337 | 0.266 | 0.293 | 2.263 | 0.914 |
| | | | | | | 720 | 0.490 | 0.478 | 0.426 | 0.445 | 0.448 | 0.440 | 0.394 | 0.396 | 0.350 | 0.347 | 2.119 | 0.958 |
| | | | | | | Avg | 0.438 | 0.434 | 0.373 | 0.402 | 0.382 | 0.398 | 0.273 | 0.321 | 0.246 | 0.274 | 2.349 | 0.951 |

*Table 11.* Full ablation results on expert diversity and the expert relation layer across six datasets. We evaluate the impact of the heterogeneous expert pool (Model ①) versus the removal of the cyclic relation layer (Model ②) and various homogeneous configurations: All Identity (Model ③), All Trend (Model ④), All Seasonality (Model ⑤), and All Fluctuation (Model ⑥). *Note: For ILI dataset, the prediction lengths are 24, 36, 48, and 60, corresponding to the shown 96, 192, 336, and 720 positions respectively.*

| Model | Experts Diversity | | | | | Relation Layer | Len. | ETTh1 | | ETTh2 | | ETTm1 | | ETTm2 | | Weather | | ILI | |
|---|---|---|---|---|---|---|---|---|---|---|---|---|---|---|---|---|---|---|---|
| | Hetero-geneous | All identity | All trend | All seasonality | All fluctuation | | | MSE | MAE | MSE | MAE | MSE | MAE | MSE | MAE | MSE | MAE | MSE | MAE |
| ① | ✓ | | | | | ✓ | 96 | 0.369 | 0.393 | 0.283 | 0.335 | 0.311 | 0.352 | 0.168 | 0.253 | 0.154 | 0.201 | 1.930 | 0.873 |
| | | | | | | | 192 | 0.417 | 0.420 | 0.358 | 0.388 | 0.358 | 0.381 | 0.232 | 0.296 | 0.202 | 0.246 | 2.143 | 0.912 |
| | | | | | | | 336 | 0.459 | 0.445 | 0.408 | 0.427 | 0.388 | 0.403 | 0.290 | 0.334 | 0.260 | 0.289 | 1.914 | 0.870 |
| | | | | | | | 720 | 0.471 | 0.463 | 0.422 | 0.442 | 0.447 | 0.441 | 0.388 | 0.391 | 0.343 | 0.344 | 1.939 | 0.895 |
| | | | | | | | Avg | **0.429** | **0.430** | **0.368** | **0.398** | **0.376** | **0.394** | **0.269** | **0.318** | **0.240** | **0.270** | **1.981** | **0.888** |
| ② | ✓ | | | | | ✗ | 96 | 0.375 | 0.393 | 0.290 | 0.341 | 0.320 | 0.358 | 0.174 | 0.257 | 0.163 | 0.209 | 2.455 | 0.937 |
| | | | | | | | 192 | 0.419 | 0.421 | 0.364 | 0.393 | 0.361 | 0.383 | 0.237 | 0.299 | 0.209 | 0.251 | 2.299 | 0.925 |
| | | | | | | | 336 | 0.487 | 0.452 | 0.426 | 0.438 | 0.389 | 0.404 | 0.296 | 0.338 | 0.266 | 0.292 | 2.351 | 0.951 |
| | | | | | | | 720 | 0.482 | 0.469 | 0.424 | 0.444 | 0.452 | 0.441 | 0.389 | 0.393 | 0.349 | 0.347 | 2.150 | 0.925 |
| | | | | | | | Avg | 0.441 | 0.434 | 0.376 | 0.404 | 0.381 | 0.396 | 0.274 | 0.322 | 0.247 | 0.275 | 2.314 | 0.934 |
| ③ | | ✓ | | | | ✓ | 96 | 0.375 | 0.393 | 0.292 | 0.341 | 0.323 | 0.361 | 0.173 | 0.257 | 0.162 | 0.207 | 2.531 | 0.988 |
| | | | | | | | 192 | 0.420 | 0.420 | 0.362 | 0.391 | 0.362 | 0.383 | 0.237 | 0.299 | 0.208 | 0.250 | 2.494 | 0.992 |
| | | | | | | | 336 | 0.484 | 0.452 | 0.427 | 0.441 | 0.395 | 0.408 | 0.295 | 0.337 | 0.268 | 0.294 | 2.433 | 0.985 |
| | | | | | | | 720 | 0.480 | 0.469 | 0.430 | 0.447 | 0.455 | 0.443 | 0.389 | 0.393 | 0.347 | 0.346 | 2.261 | 0.974 |
| | | | | | | | Avg | 0.440 | 0.434 | 0.378 | 0.405 | 0.384 | 0.399 | 0.274 | 0.322 | 0.246 | 0.274 | 2.430 | 0.985 |
| ④ | | | ✓ | | | ✓ | 96 | 0.377 | 0.395 | 0.291 | 0.342 | 0.329 | 0.367 | 0.173 | 0.256 | 0.162 | 0.207 | 2.480 | 0.984 |
| | | | | | | | 192 | 0.421 | 0.420 | 0.365 | 0.390 | 0.360 | 0.381 | 0.237 | 0.299 | 0.209 | 0.251 | 2.455 | 0.989 |
| | | | | | | | 336 | 0.479 | 0.448 | 0.414 | 0.432 | 0.396 | 0.410 | 0.295 | 0.337 | 0.267 | 0.293 | 2.404 | 0.979 |
| | | | | | | | 720 | 0.494 | 0.479 | 0.425 | 0.444 | 0.457 | 0.445 | 0.389 | 0.392 | 0.347 | 0.345 | 2.295 | 0.978 |
| | | | | | | | Avg | 0.443 | 0.435 | 0.374 | 0.402 | 0.386 | 0.401 | 0.273 | 0.321 | 0.246 | 0.274 | 2.409 | 0.982 |
| ⑤ | | | | ✓ | | ✓ | 96 | 0.378 | 0.393 | 0.289 | 0.339 | 0.324 | 0.361 | 0.173 | 0.257 | 0.162 | 0.208 | 1.793 | 0.827 |
| | | | | | | | 192 | 0.426 | 0.421 | 0.376 | 0.397 | 0.362 | 0.383 | 0.236 | 0.299 | 0.209 | 0.251 | 2.353 | 0.979 |
| | | | | | | | 336 | 0.474 | 0.443 | 0.411 | 0.429 | 0.393 | 0.407 | 0.296 | 0.338 | 0.265 | 0.291 | 1.985 | 0.867 |
| | | | | | | | 720 | 0.474 | 0.465 | 0.430 | 0.448 | 0.455 | 0.444 | 0.392 | 0.394 | 0.347 | 0.345 | 2.152 | 0.944 |
| | | | | | | | Avg | 0.438 | 0.431 | 0.377 | 0.403 | 0.384 | 0.399 | 0.274 | 0.322 | 0.246 | 0.274 | 2.071 | 0.904 |
| ⑥ | | | | | ✓ | ✓ | 96 | 0.374 | 0.392 | 0.291 | 0.340 | 0.329 | 0.367 | 0.173 | 0.256 | 0.162 | 0.207 | 2.058 | 0.858 |
| | | | | | | | 192 | 0.421 | 0.419 | 0.372 | 0.394 | 0.363 | 0.382 | 0.240 | 0.300 | 0.209 | 0.251 | 2.531 | 0.945 |
| | | | | | | | 336 | 0.482 | 0.448 | 0.412 | 0.429 | 0.394 | 0.406 | 0.297 | 0.338 | 0.265 | 0.292 | 2.275 | 0.924 |
| | | | | | | | 720 | 0.476 | 0.466 | 0.442 | 0.455 | 0.454 | 0.442 | 0.390 | 0.393 | 0.348 | 0.346 | 2.153 | 0.949 |
| | | | | | | | Avg | 0.438 | 0.431 | 0.379 | 0.404 | 0.385 | 0.399 | 0.275 | 0.322 | 0.246 | 0.274 | 2.254 | 0.919 |

*Table 12.* Full ablation results on adaptation strategies across seven datasets. We analyze the contribution of each component in the adaptive strategies: comparing MMD-based drift detection against simple statistical detection (Model ②), validating the Post-Addition Alignment against fine-tuning base experts (Model ③), evaluating the Drift Pattern Profiler against random new expert selection (Model ④), and assessing the necessity of expert pruning (Model ⑤). *Note: For ILI dataset, the prediction lengths are 24, 36, 48, and 60, corresponding to the shown 96, 192, 336, and 720 positions respectively.*

| Model | Drift Detection | | New Expert Alignment Strategy | | New Expert Selection | | Experts Pruning | Len. | ETTh1 | | ETTh2 | | ETTm1 | | ETTm2 | | Weather | | ILI | | Exchange | |
|---|---|---|---|---|---|---|---|---|---|---|---|---|---|---|---|---|---|---|---|---|---|---|
| | MMD | Simple | Post-Addition Alignment | Fine-tune the Base Experts | Drift Pattern Profiler | Random | | | MSE | MAE | MSE | MAE | MSE | MAE | MSE | MAE | MSE | MAE | MSE | MAE | MSE | MAE |
| ① | ✓ | | ✓ | | ✓ | | ✓ | 96 | 0.369 | 0.393 | 0.283 | 0.335 | 0.311 | 0.352 | 0.168 | 0.253 | 0.154 | 0.201 | 1.930 | 0.873 | 0.080 | 0.197 |
| | | | | | | | | 192 | 0.417 | 0.420 | 0.358 | 0.388 | 0.358 | 0.381 | 0.232 | 0.296 | 0.202 | 0.246 | 2.143 | 0.912 | 0.170 | 0.293 |
| | | | | | | | | 336 | 0.459 | 0.445 | 0.408 | 0.427 | 0.388 | 0.403 | 0.290 | 0.334 | 0.260 | 0.289 | 1.914 | 0.870 | 0.325 | 0.412 |
| | | | | | | | | 720 | 0.471 | 0.463 | 0.422 | 0.442 | 0.447 | 0.441 | 0.388 | 0.391 | 0.343 | 0.344 | 1.939 | 0.895 | 0.828 | 0.686 |
| | | | | | | | | Avg | **0.429** | **0.430** | **0.368** | **0.398** | **0.376** | **0.394** | **0.269** | **0.318** | **0.240** | **0.270** | **1.981** | **0.888** | **0.351** | **0.397** |
| ② | | ✓ | ✓ | | ✓ | | ✓ | 96 | 0.373 | 0.392 | 0.284 | 0.337 | 0.324 | 0.361 | 0.171 | 0.255 | 0.161 | 0.206 | 1.853 | 0.834 | 0.082 | 0.199 |
| | | | | | | | | 192 | 0.416 | 0.419 | 0.364 | 0.393 | 0.359 | 0.381 | 0.238 | 0.300 | 0.209 | 0.250 | 2.408 | 0.984 | 0.172 | 0.294 |
| | | | | | | | | 336 | 0.477 | 0.453 | 0.415 | 0.432 | 0.396 | 0.407 | 0.295 | 0.337 | 0.265 | 0.291 | 2.109 | 0.903 | 0.362 | 0.436 |
| | | | | | | | | 720 | 0.479 | 0.468 | 0.422 | 0.442 | 0.451 | 0.441 | 0.393 | 0.396 | 0.349 | 0.347 | 1.960 | 0.893 | 0.910 | 0.721 |
| | | | | | | | | Avg | 0.436 | 0.433 | 0.371 | 0.401 | 0.382 | 0.397 | 0.274 | 0.322 | 0.246 | 0.274 | 2.082 | 0.904 | 0.382 | 0.413 |
| ③ | ✓ | | | ✓ | ✓ | | ✓ | 96 | 0.371 | 0.393 | 0.285 | 0.337 | 0.319 | 0.357 | 0.172 | 0.256 | 0.164 | 0.210 | 1.936 | 0.838 | 0.082 | 0.199 |
| | | | | | | | | 192 | 0.416 | 0.421 | 0.362 | 0.391 | 0.363 | 0.384 | 0.238 | 0.300 | 0.209 | 0.251 | 2.465 | 0.991 | 0.172 | 0.294 |
| | | | | | | | | 336 | 0.459 | 0.445 | 0.412 | 0.428 | 0.390 | 0.403 | 0.295 | 0.337 | 0.267 | 0.293 | 2.387 | 0.974 | 0.329 | 0.414 |
| | | | | | | | | 720 | 0.473 | 0.464 | 0.425 | 0.444 | 0.452 | 0.442 | 0.392 | 0.394 | 0.346 | 0.344 | 2.164 | 0.948 | 0.819 | 0.682 |
| | | | | | | | | Avg | 0.430 | 0.431 | 0.371 | 0.400 | 0.381 | 0.397 | 0.274 | 0.322 | 0.246 | 0.274 | 2.238 | 0.938 | 0.351 | 0.397 |
| ④ | ✓ | | ✓ | | | ✓ | ✓ | 96 | 0.371 | 0.393 | 0.287 | 0.339 | 0.322 | 0.360 | 0.176 | 0.260 | 0.155 | 0.201 | 1.942 | 0.863 | 0.082 | 0.199 |
| | | | | | | | | 192 | 0.416 | 0.421 | 0.362 | 0.391 | 0.363 | 0.384 | 0.237 | 0.299 | 0.206 | 0.249 | 2.266 | 0.944 | 0.172 | 0.294 |
| | | | | | | | | 336 | 0.459 | 0.445 | 0.412 | 0.428 | 0.390 | 0.403 | 0.295 | 0.338 | 0.265 | 0.292 | 2.672 | 1.055 | 0.329 | 0.414 |
| | | | | | | | | 720 | 0.472 | 0.463 | 0.422 | 0.442 | 0.450 | 0.440 | 0.391 | 0.393 | 0.347 | 0.346 | 2.162 | 0.947 | 0.877 | 0.710 |
| | | | | | | | | Avg | 0.429 | 0.431 | 0.371 | 0.400 | 0.381 | 0.397 | 0.275 | 0.322 | 0.243 | 0.272 | 2.260 | 0.952 | 0.365 | 0.404 |
| ⑤ | ✓ | | ✓ | | ✓ | | × | 96 | 0.374 | 0.393 | 0.293 | 0.343 | 0.332 | 0.367 | 0.171 | 0.255 | 0.157 | 0.203 | 2.097 | 0.875 | 0.082 | 0.199 |
| | | | | | | | | 192 | 0.417 | 0.420 | 0.367 | 0.392 | 0.362 | 0.383 | 0.233 | 0.297 | 0.206 | 0.249 | 2.384 | 0.935 | 0.175 | 0.298 |
| | | | | | | | | 336 | 0.466 | 0.445 | 0.416 | 0.435 | 0.393 | 0.406 | 0.293 | 0.335 | 0.266 | 0.292 | 2.277 | 0.915 | 0.325 | 0.415 |
| | | | | | | | | 720 | 0.490 | 0.479 | 0.423 | 0.443 | 0.454 | 0.442 | 0.393 | 0.394 | 0.350 | 0.348 | 2.281 | 0.981 | 0.840 | 0.689 |
| | | | | | | | | Avg | 0.437 | 0.434 | 0.375 | 0.403 | 0.385 | 0.400 | 0.272 | 0.320 | 0.245 | 0.273 | 2.260 | 0.927 | 0.355 | 0.399 |

